# Tackling the Objective Inconsistency Problem in Heterogeneous Federated Optimization

**Jianyu Wang**
Carnegie Mellon University
Pittsburgh, PA 15213
jianyuw1@andrew.cmu.edu

**Qinghua Liu**
Princeton University
Princeton, NJ 08544
qinghual@princeton.edu

**Hao Liang**
Carnegie Mellon University
Pittsburgh, PA 15213
hliang2@andrew.cmu.edu

**Gauri Joshi**
Carnegie Mellon University
Pittsburgh, PA 15213
gaurij@andrew.cmu.edu

**H. Vincent Poor**
Princeton University
Princeton, NJ 08544
poor@princeton.edu

## Abstract

In federated learning, heterogeneity in the clients' local datasets and computation speeds results in large variations in the number of local updates performed by each client in each communication round. Naive weighted aggregation of such models causes objective inconsistency, that is, the global model converges to a stationary point of a mismatched objective function which can be arbitrarily different from the true objective. This paper provides a general framework to analyze the convergence of heterogeneous federated optimization algorithms. It subsumes previously proposed methods such as FedAvg and FedProx, and provides the first principled understanding of the solution bias and the convergence slowdown due to objective inconsistency. Using insights from this analysis, we propose FedNova, a normalized averaging method that eliminates objective inconsistency while preserving fast error convergence.

## 1 Introduction

Federated learning [1–5] is an emerging sub-area of distributed optimization where both data collection and model training is pushed to a large number of edge clients that have limited communication and computation capabilities. Unlike traditional distributed optimization [6, 7] where consensus (either through a central server or peer-to-peer communication) is performed after every local gradient computation, in federated learning, the subset of clients selected in each communication round perform multiple local updates before these models are aggregated in order to update a global model.

**Heterogeneity in the Number of Local Updates in Federated Learning.** The clients participating in federated learning are typically highly heterogeneous, both in the size of their local datasets as well as their computation speeds. The original paper on federated learning [1] proposed that each client performs $E$ *epochs* (traversals of their local dataset) of local-update stochastic gradient descent (SGD) with a mini-batch size $B$. Thus, if a client has $n_i$ local data samples, the number of local SGD iterations is $\tau_i = \lfloor En_i/B \rfloor$, which can vary widely across clients. The heterogeneity in the number of local SGD iterations is exacerbated by relative variations in the clients' computing speeds. Within a given wall-clock time interval, faster clients can perform more local updates than slower clients. The number of local updates made by a client can also vary across communication rounds due to unpredictable straggling or slowdown caused by background processes, outages, memory limitations etc. Finally, clients may use different learning rates and local solvers (instead of vanilla

SGD, they may use proximal gradient methods or adaptive learning rate schedules) which may result in heterogeneity in the model progress at each client.

**Heterogeneity in Local Updates Causes Objective Inconsistency.** Most recent works that analyze the convergence of federated optimization algorithms [8–37] assume that number of local updates is the same across all clients (that is, $\tau_i = \tau$ for all clients $i$). These works show that periodic consensus between the locally trained client models attains a stationary point of the global objective function $F(\boldsymbol{x}) = \sum_{i=1}^{m} n_i F_i(\boldsymbol{x})/n$, which is a sum of local objectives weighted by the dataset size $n_i$. However, no current analysis provides insight into the convergence of local-update or federated optimization algorithms in the practical setting when the number of local updates $\tau_i$ varies across clients $1, \ldots, m$. In fact, as we show in Section 3, *standard averaging of client models after heterogeneous local updates results in convergence to a stationary point – not of the original objective function $F(\boldsymbol{x})$, but of an inconsistent objective $\widetilde{F}(\boldsymbol{x})$, which can be arbitrarily different from $F(\boldsymbol{x})$ depending upon the relative values of $\tau_i$.* To gain intuition into this phenomenon, observe in Figure 1 that if client 1 performs more local updates, then the updated $\boldsymbol{x}^{(t+1,0)}$ strays towards the local minimum $\boldsymbol{x}_1^*$, away from the true global minimum $\boldsymbol{x}^*$.

**The Need for a General Analysis Framework.** A naive approach to overcome heterogeneity is to fix a target number of local updates $\tau$ that each client must finish within a communication round and keep fast nodes idle while the slow clients finish their updates. This method will ensure objective consistency (that is, the surrogate objective $\widetilde{F}(\boldsymbol{x})$ equals to the true objective $F(\boldsymbol{x})$), nonetheless, waiting for the slowest one can significantly increase the total training time. More sophisticated approaches such as `FedProx` [38], `VRLSGD` [21] and `SCAFFOLD` [20], designed to handle non-IID local datasets, can be used to reduce (not eliminate) objective inconsistency to some extent, but these methods either result in slower convergence or require additional communication and memory. So far, there is no rigorous understanding of the objective inconsistency and the speed of convergence for this challenging setting of federated learning with heterogeneous local updates. It is also unclear how to best combine models trained with heterogeneous levels of local progress.

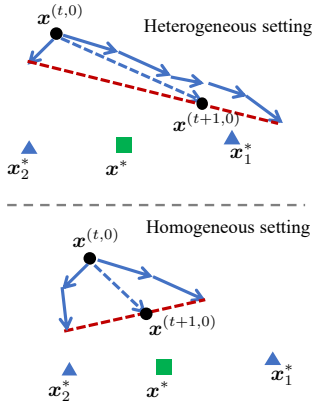

Figure 1: Model updates in the parameter space. Green squares and blue triangles denote the minima of global and local objectives, respectively.

**Contributions of this Paper.** To the best of our knowledge, this work provides the first fundamental understanding of the bias in the solution (caused by objective inconsistency) and how the convergence rate is influenced by heterogeneity in clients' local progress. In Section 4 we propose a general theoretical framework that allows heterogeneous number of local updates, non-IID local datasets as well as different local solvers such as GD, SGD, SGD with proximal gradients, gradient tracking, adaptive learning rates, momentum, etc. It subsumes existing methods such as `FedAvg` and `FedProx` and provides novel insights on their convergence behaviors. In Section 5 we propose `FedNova`, a method that correctly weigh local models when averaging. It ensures objective consistency while preserving fast error convergence and outperforms existing methods as shown in Section 6. `FedNova` works with any local solver and server optimizer and is therefore complementary to existing approaches such as [38, 39, 20, 40].

## 2 System Model and Prior Work

**The Federated Heterogeneous Optimization Setting.** In federated learning, a total of $m$ clients aim to jointly solve the following optimization problem:

$$\min_{\boldsymbol{x} \in \mathbb{R}^d} \left[ F(\boldsymbol{x}) := \sum_{i=1}^{m} p_i F_i(\boldsymbol{x}) \right] \tag{1}$$

where $p_i = n_i/n$ denotes the relative sample size, and $F_i(\boldsymbol{x}) = \frac{1}{n_i} \sum_{\xi \in \mathcal{D}_i} f_i(\boldsymbol{x}; \xi)$ is the local objective function at the $i$-th client. Here, $f_i$ is the loss function (possibly non-convex) defined by the learning model and $\xi$ represents a data sample from local dataset $\mathcal{D}_i$. In the $t$-th communication

round, each client independently runs $\tau_i$ iterations of local solver (*e.g.*, SGD) starting from the current global model $\boldsymbol{x}^{(t,0)}$ to optimize its own local objective.

In our theoretical framework, we treat $\tau_i$ as an arbitrary scalar which can also vary across rounds. In practice, if clients run for the same local epochs $E$, then $\tau_i = \lfloor En_i/B \rfloor$, where $B$ is the mini-batch size. Alternately, if each communication round has a fixed length in terms of wall-clock time, then $\tau_i$ represents the local iterations completed by client $i$ within the time window and may change across clients (depending on their computation speeds and availability) and across communication rounds.

**The FedAvg Baseline Algorithm.** *Federated Averaging* (`FedAvg`) [1] is the first and most common algorithm used to aggregate these locally trained models at the central server at the end of each communication round. The shared global model is updated as follows:

$$\texttt{FedAvg:} \quad \boldsymbol{x}^{(t+1,0)} - \boldsymbol{x}^{(t,0)} = \sum_{i=1}^m p_i \Delta_i^{(t)} = -\sum_{i=1}^m p_i \cdot \eta \sum_{k=0}^{\tau_i-1} g_i(\boldsymbol{x}_i^{(t,k)}) \tag{2}$$

where $\boldsymbol{x}_i^{(t,k)}$ denotes client $i$'s model after the $k$-th local update in the $t$-th communication round and $\Delta_i^{(t)} = \boldsymbol{x}_i^{(t,\tau_i)} - \boldsymbol{x}_i^{(t,0)}$ denotes the cumulative local progress made by client $i$ at round $t$. Also, $\eta$ is the client learning rate and $g_i$ represents the stochastic gradient over a mini-batch of $B$ samples. When the number of clients $m$ is large, then the central server may only randomly select a subset of clients to perform computation at each round.

**Convergence Analysis of FedAvg.** [8–10] first analyze `FedAvg` by assuming the local objectives are identical and show that `FedAvg` is guaranteed to converge to a stationary point of $F(\boldsymbol{x})$. This analysis was further expanded to the non-IID data partition and client sampling cases by [11–18, 23, 24]. However, in all these works, they assume that the number of local steps and the client optimizer are the same across all clients. Besides, asynchronous federated optimization algorithms proposed in [41, 9] take a different approach of allowing clients make updates to stale versions of the global model, and their analyses are limited to IID local datasets and convex local functions.

**FedProx: Improving FedAvg by Adding a Proximal Term.** To alleviate inconsistency due to non-IID data and heterogeneous local updates, [38] proposes adding a proximal term $\frac{\mu}{2}\|\boldsymbol{x} - \boldsymbol{x}^{(t,0)}\|^2$ to each local objective, where $\mu \geq 0$ is a tunable parameter. This proximal term pulls each local model backward closer to the global model $\boldsymbol{x}^{(t,0)}$. Although [38] empirically shows that `FedProx` improves `FedAvg`, its convergence analysis is limited by assumptions that are stronger than previous `FedAvg` analysis and only works for sufficiently large $\mu$. Since `FedProx` is a special case of our general framework, our convergence analysis provides sharp insights into the effect of $\mu$. We show that a larger $\mu$ mitigates (but does not eliminate) objective inconsistency, albeit at an expense of slower convergence. Our proposed `FedNova` method can improve `FedProx` by guaranteeing consistency without slowing down convergence.

**Improving FedAvg via Momentum and Cross-client Variance Reduction.** The performance of `FedAvg` has been improved in recent literature by applying momentum on the server side [25, 42, 40], or using cross-client variance reduction such as `VRLSGD` and `SCAFFOLD` [21, 20]. Again, these works do not consider heterogeneous local progress. Our proposed normalized averaging method `FedNova` is orthogonal to and can be easily combined with these acceleration or variance-reduction techniques. Moreover, `FedNova` is also compatible with and complementary to gradient compression/quantization [43–48] and fair aggregation techniques [49, 50].

## 3 A Case Study to Demonstrate the Objective Inconsistency Problem

In this section, we use a simple quadratic model to illustrate the convergence problem. Suppose that the local objective functions are $F_i(\boldsymbol{x}) = \frac{1}{2}\|\boldsymbol{x} - \boldsymbol{e}_i\|^2$, where $\boldsymbol{e}_i \in \mathbb{R}^d$ is an arbitrary vector and it is the minimum $\boldsymbol{x}_i^*$ of the local objective. Consider that the global objective function is defined as

$$F(\boldsymbol{x}) = \frac{1}{m}\sum_{i=1}^m F_i(\boldsymbol{x}) = \sum_{i=1}^m \frac{1}{2}\|\boldsymbol{x} - \boldsymbol{e}_i\|^2, \quad \text{which is minimized by } \boldsymbol{x}^* = \frac{1}{m}\sum_{i=1}^m \boldsymbol{e}_i. \tag{3}$$

Below, we show that the convergence point of `FedAvg` can be arbitrarily away from $\boldsymbol{x}^*$.

**Lemma 1** (**Objective Inconsistency in FedAvg**). *For the objective function in* (3)*, if client $i$ performs $\tau_i$ local steps per round, then* **FedAvg** *(with sufficiently small learning rate $\eta$, deterministic gradients and full client participation) will converge to*

$$\tilde{\boldsymbol{x}}_{\texttt{FedAvg}}^* = \lim_{T\to\infty} \boldsymbol{x}^{(T,0)} = \frac{\sum_{i=1}^m \tau_i \boldsymbol{e}_i}{\sum_{i=1}^m \tau_i}, \text{which minimizes the surrogate obj.:} \widetilde{F}(\boldsymbol{x}) = \frac{\sum_{i=1}^m \tau_i F_i(\boldsymbol{x})}{\sum_{i=1}^m \tau_i}.$$

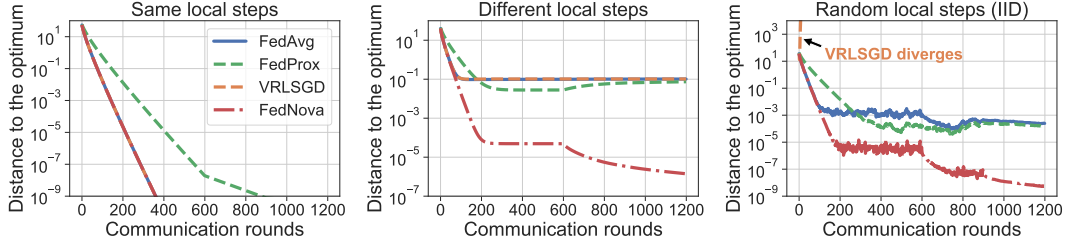

Figure 2: Simulations comparing the `FedAvg`, `FedProx` ($\mu = 1$), `VRLSGD` and our proposed `FedNova` algorithms for 30 clients with the quadratic objectives defined in (3), where $e_i \sim \mathcal{N}(0, 0.01\boldsymbol{I})$, $i \in [1, 30]$. Clients perform GD with $\eta = 0.05$, which is decayed by a factor of 5 at rounds 600 and 900. *Left*: Clients perform the same number of local steps $\tau_i = 30$ – `FedNova` is equivalent to `FedAvg` in this case; *Middle*: Clients take different local steps $\tau_i \in [1, 96]$ with mean 30 but fixed across rounds; *Right*: local steps are IID, and time-varying Gaussians with mean 30, *i.e.*, $\tau_i(t) \in [1, 96]$. `FedNova` significantly outperforms others in the heterogeneous $\tau_i$ setting.

The proof (of a more general version of Lemma 1) is deferred to the Appendix. While `FedAvg` aims at optimizing $F(\boldsymbol{x})$, it actually converges to the optimum of a surrogate objective $\widetilde{F}(\boldsymbol{x})$. As illustrated in Figure 2, there can be an arbitrarily large gap between $\tilde{\boldsymbol{x}}^*_{\texttt{FedAvg}}$ and $\boldsymbol{x}^*$ depending on the relative values of $\tau_i$ and $F_i(\boldsymbol{x})$. This non-vanishing gap also occurs when the local steps $\tau_i$ are IID random variables across clients and communication rounds (see the right panel in Figure 2).

**Convergence Problem in Other Federated Algorithms.** We can generalize Lemma 1 to the case of `FedProx` to demonstrate its convergence gap, as given in the Appendix. From the simulations shown in Figure 2, observe that `FedProx` can slightly improve on the optimality gap of `FedAvg`, but it converges slower. Besides, previous cross-client variance reduction methods such as variance-reduced local SGD (`VRLSGD`) [21] and `SCAFFOLD` [20] are only designed for homogeneous local steps case. In the considered heterogeneous setting, if we replace the same local steps $\tau$ in `VRLSGD` by different $\tau_i$'s, then we observe that it has drastically different convergence under different settings and even diverge when clients perform random local steps (see the right panel in Figure 2). These observations emphasize the critical need for a deeper understanding of objective inconsistency and new heterogeneous federated optimization algorithms.

## 4 New Theoretical Framework For Heterogeneous Federated Optimization

We now present a general theoretical framework that subsumes a suite of federated optimization algorithms and helps analyze the effect of objective inconsistency on their error convergence. Although the results are presented for the full client participation setting, it is fairly easy to extend them to the case where a subset of clients are randomly sampled in each round[1].

### 4.1 A Generalized Update Rule for Heterogeneous Federated Optimization

Recall from (2) that the update rule of federated optimization algorithms can be written as $\boldsymbol{x}^{(t+1,0)} - \boldsymbol{x}^{(t,0)} = \sum_{i=1}^{m} p_i \Delta_i^{(t)}$, where $\Delta_i^{(t)} := \boldsymbol{x}^{(t,\tau_i)} - \boldsymbol{x}^{(t,0)}$ denote the local parameter changes of client $i$ at round $t$ and $p_i = n_i/n$, the fraction of data at client $i$. We re-write this update rule in a more general form as follows:

$$\boldsymbol{x}^{(t+1,0)} - \boldsymbol{x}^{(t,0)} = -\tau_{\text{eff}} \sum_{i=1}^{m} w_i \cdot \eta \boldsymbol{d}_i^{(t)}, \quad \text{which optimizes } \widetilde{F}(\boldsymbol{x}) = \sum_{i=1}^{m} w_i F_i(\boldsymbol{x}). \quad (4)$$

The following three key elements of this update rule take different forms for different algorithms:

1. **Locally averaged gradient** $d_i^{(t)}$: Without loss of generality, we can rewrite the cumulative local changes as $\Delta_i^{(t)} = -\eta G_i^{(t)} a_i$, where $G_i^{(t)} = [g_i(x_i^{(t,0)}), g_i(x_i^{(t,1)}), \dots, g_i(x_i^{(t,\tau_i)})] \in \mathbb{R}^{d \times \tau_i}$ stacks all stochastic gradients in the $t$-th round, and $a_i \in \mathbb{R}^{\tau_i}$ is a non-negative vector and defines how stochastic gradients are locally accumulated. Then, by normalizing the gradient weights $a_i$, the locally averaged gradient is defined as $d_i^{(t)} = G_i^{(t)} a_i / \|a_i\|_1$. The normalizing factor $\|a_i\|_1$ in the denominator is the $\ell_1$ norm of the vector $a_i$. By setting different $a_i$, (4) works for most common client optimizers such as SGD with proximal updates, local momentum, and variable learning rate, and more generally, any solver whose cumulative changes $\Delta_i^{(t)} = -\eta G_i^{(t)} a_i$, a linear combination of local gradients.

Specifically, if the client optimizer is vanilla SGD (*i.e.*, the case of FedAvg), then $a_i = [1, 1, \dots, 1] \in \mathbb{R}^{\tau_i}$ and $\|a_i\|_1 = \tau_i$. As a result, the normalized gradient is just a simple average of all stochastic gradients within current round: $d_i^{(t)} = G_i^{(t)} a_i / \tau_i = \sum_{k=0}^{\tau_i - 1} g_i(x_i^{(t,k)}) / \tau_i$. Later in this section, we will present more specific examples on how to set $a_i$ in other algorithms.

2. **Aggregation weights** $w_i$: Each client's locally averaged gradient $d_i$ is multiplied with weight $w_i$ when computing the aggregated gradient $\sum_{i=1}^m w_i d_i$. By definition, these weights satisfy $\sum_{i=1}^m w_i = 1$. Observe that these weights determine the surrogate objective $\widetilde{F}(x) = \sum_{i=1}^m w_i F_i(x)$, which is optimized by the general algorithm in (4) instead of the true global objective $F(x) = \sum_{i=1}^m p_i F_i(x)$ – we will prove this formally in Theorem 1.

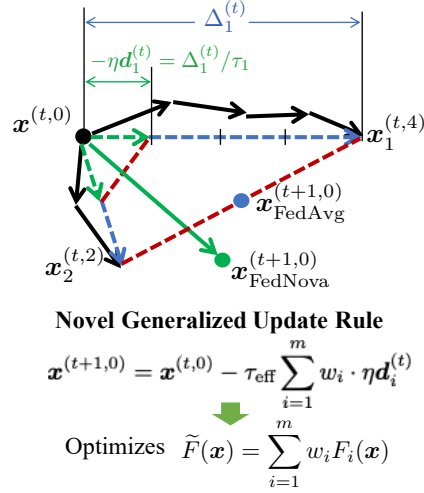

**Novel Generalized Update Rule**

$$x^{(t+1,0)} = x^{(t,0)} - \tau_{\text{eff}} \sum_{i=1}^m w_i \cdot \eta d_i^{(t)}$$

Optimizes $\widetilde{F}(x) = \sum_{i=1}^m w_i F_i(x)$

Figure 3: Comparison between the novel framework and FedAvg in the model parameter space. Solid black arrows denote local updates at clients. Green and blue dots denote the global updates made by the novel generalized update rule and FedAvg respectively. While $w_i$ controls the direction of the solid green arrow, effective steps $\tau_{\text{eff}}$ determines how far the global model moves along with this direction. FedAvg implicitly assigns too higher weights for clients with more local steps, resulting in a biased global direction.

3. **Effective number of steps** $\tau_{\text{eff}}$: Since client $i$ makes $\tau_i$ local updates, the average number of local SGD steps per communication round is $\bar{\tau} = \sum_{i=1}^m \tau_i / m$. However, the server can scale up or scale down the effect of the aggregated updates by setting the parameter $\tau_{\text{eff}}$ larger or smaller than $\bar{\tau}$ (analogous to choosing a global learning rate [25, 40]). We refer to the ratio $\bar{\tau} / \tau_{\text{eff}}$ as the *slowdown*, and it features prominently in the convergence analysis presented in Section 4.2.

The general rule (4) enables us to freely choose $\tau_{\text{eff}}$ and $w_i$ for a given local solver $a_i$, which helps design fast and consistent algorithms such as FedNova, the normalized averaging method proposed in Section 5. In Figure 3, we further illustrate how the above key elements influence the algorithm and compare the novel generalized update rule and FedAvg in the model parameter space. Besides, in terms of the implementation, the server is not necessary to know the specific form of local accumulation vector $a_i$. Each client can send the normalized update $-\eta d_i^{(t)}$ to the central server, which is just a re-scaled version of cumulative local changes $\Delta_i^{(t)}$.

**Previous Algorithms as Special Cases.** Any previous algorithms whose cumulative changes $\Delta_i^{(t)} = -\eta G_i^{(t)} a_i$, a linear combination of local gradients can be subsumed by the above formulation. One can validate this as follows:

$$x^{(t+1,0)} - x^{(t,0)} = \sum_{i=1}^m p_i \Delta_i^{(t)} = -\sum_{i=1}^m p_i \|a_i\|_1 \cdot \frac{\eta G_i^{(t)} a_i}{\|a_i\|_1} \tag{5}$$

$$= -\underbrace{\left( \sum_{i=1}^m p_i \|a_i\|_1 \right)}_{\tau_{\text{eff}}: \text{ effective local steps}} \sum_{i=1}^m \eta \underbrace{\left( \frac{p_i \|a_i\|_1}{\sum_{i=1}^m p_i \|a_i\|_1} \right)}_{w_i: \text{ weight}} \underbrace{\left( \frac{G_i^{(t)} a_i}{\|a_i\|_1} \right)}_{d_i: \text{ normalized gradient}}. \tag{6}$$

Unlike the more general form (4), in (6), which subsumes the following previous methods, $\tau_{\text{eff}}$ and $w_i$ are implicitly fixed by the choice of the local solver (*i.e.*, the choice of $\boldsymbol{a}_i$). Due to space limitations, the derivations of following examples are relegated to the Appendix.

• **Vanilla SGD as Local Solver (FedAvg).** In `FedAvg`, the local solver is SGD such that $\boldsymbol{a}_i = [1, 1, \ldots, 1] \in \mathbb{R}^{\tau_i}$ and $\|\boldsymbol{a}_i\|_1 = \tau_i$. As a consequence, the locally averaged gradient $\boldsymbol{d}_i$ is a simple average over $\tau_i$ iterations, $\tau_{\text{eff}} = \sum_{i=1}^m p_i \tau_i$, and $w_i = p_i \tau_i / \sum_{i=1}^m p_i \tau_i$. That is, the normalized gradients with more local steps will be implicitly assigned higher weights.

• **Proximal SGD as Local Solver (FedProx).** In `FedProx`, local SGD steps are corrected by a proximal term. It can be shown that $\boldsymbol{a}_i = [(1-\alpha)^{\tau_i-1}, (1-\alpha)^{\tau_i-2}, \ldots, (1-\alpha), 1] \in \mathbb{R}^{\tau_i}$, where $\alpha = \eta \mu$ and $\mu$ is a tunable parameter. In this case, we have $\|\boldsymbol{a}_i\|_1 = [1 - (1-\alpha)^{\tau_i}]/\alpha$ and hence,

$$\tau_{\text{eff}} = \alpha^{-1} \sum_{i=1}^m p_i [1 - (1-\alpha)^{\tau_i}], \quad w_i = p_i [1 - (1-\alpha)^{\tau_i}] / \sum_{i=1}^m p_i [1 - (1-\alpha)^{\tau_i}]. \quad (7)$$

When $\alpha = 0$, `FedProx` is equivalent to `FedAvg`. As $\alpha = \eta \mu$ increases, the $w_i$ in `FedProx` is more similar to $p_i$, thus making the surrogate objective $\widetilde{F}(\boldsymbol{x})$ more consistent. However, a larger $\alpha$ corresponds to smaller $\tau_{\text{eff}}$, which slows down convergence, as we discuss more in the next subsection.

• **SGD with Decayed Learning Rate as Local Solver.** Suppose the clients' local learning rates are exponentially decayed, then we have $\boldsymbol{a}_i = [1, \gamma_i, \ldots, \gamma_i^{\tau_i-1}]$ where $\gamma_i \geq 0$ can vary across clients. As a result, we have $\|\boldsymbol{a}_i\|_1 = (1 - \gamma_i^{\tau_i})/(1 - \gamma_i)$ and $w_i \propto p_i (1 - \gamma_i^{\tau_i})/(1 - \gamma_i)$. Comparing with the case of `FedProx` (7), changing the values of $\gamma_i$ has a similar effect as changing $(1 - \alpha)$.

• **Momentum SGD as Local Solver.** If we use momentum SGD where the local momentum buffers of active clients are reset to zero at the beginning of each round [25] due to the stateless nature of cross-device FL [2], then we have $\boldsymbol{a}_i = [1 - \rho^{\tau_i}, 1 - \rho^{\tau_i-1}, \ldots, 1 - \rho]/(1 - \rho)$, where $\rho$ is the momentum factor, and $\|\boldsymbol{a}_i\|_1 = [\tau_i - \rho(1 - \rho^{\tau_i})/(1 - \rho)]/(1 - \rho)$.

More generally, the new formulation (6) suggests that $w_i \neq p_i$ whenever clients have different $\|\boldsymbol{a}_i\|_1$, which may be caused by imbalanced local updates (*i.e.*, $\boldsymbol{a}_i$'s have different dimensions), or various local learning rate/momentum schedules (*i.e.*, $\boldsymbol{a}_i$'s have different scales).

## 4.2 Convergence Analysis for Smooth Non-Convex Functions

In Theorem 1 and Theorem 2 below we provide a convergence analysis for the general update rule (4) and quantify the solution bias due to objective inconsistency. The analysis relies on Assumptions 1 and 2 used in the standard analysis of SGD [51] and Assumption 3 commonly used in the federated optimization literature [38, 12, 13, 20, 40, 2] to capture the dissimilarities of local objectives.

**Assumption 1** (Smoothness). *Each local objective function is Lipschitz smooth, that is,* $\|\nabla F_i(\boldsymbol{x}) - \nabla F_i(\boldsymbol{y})\| \leq L \|\boldsymbol{x} - \boldsymbol{y}\|, \forall i \in \{1, 2, \ldots, m\}$.

**Assumption 2** (Unbiased Gradient and Bounded Variance). *The stochastic gradient at each client is an unbiased estimator of the local gradient:* $\mathbb{E}_\xi[g_i(\boldsymbol{x}|\xi)] = \nabla F_i(\boldsymbol{x})$*, and has bounded variance* $\mathbb{E}_\xi[\|g_i(\boldsymbol{x}|\xi) - \nabla F_i(\boldsymbol{x})\|^2] \leq \sigma^2, \forall i \in \{1, 2, \ldots, m\}, \sigma^2 \geq 0$.

**Assumption 3** (Bounded Dissimilarity). *For any sets of weights* $\{w_i \geq 0\}_{i=1}^m, \sum_{i=1}^m w_i = 1$*, there exist constants* $\beta^2 \geq 1, \kappa^2 \geq 0$ *such that* $\sum_{i=1}^m w_i \|\nabla F_i(\boldsymbol{x})\|^2 \leq \beta^2 \|\sum_{i=1}^m w_i \nabla F_i(\boldsymbol{x})\|^2 + \kappa^2$*. If local functions are identical to each other, then we have* $\beta^2 = 1, \kappa^2 = 0$.

**Theorem 1** (**Convergence to the Surrogate Objective** $\widetilde{F}(\boldsymbol{x})$**'s Stationary Point**). *Under Assumptions 1 to 3, any federated optimization algorithm that follows the update rule (4), will converge to a stationary point of a surrogate objective* $\widetilde{F}(\boldsymbol{x}) = \sum_{i=1}^m w_i F_i(\boldsymbol{x})$*. More specifically, if the total communication rounds $T$ is pre-determined and the learning rate $\eta$ is small enough* $\eta = \sqrt{m/\overline{\tau}T}$ *where* $\overline{\tau} = \frac{1}{m} \sum_{i=1}^m \tau_i$*, then the optimization error will be bounded as follows:*

$$\min_{t \in [T]} \mathbb{E}\|\nabla \widetilde{F}(\boldsymbol{x}^{(t,0)})\|^2 \leq \underbrace{\mathcal{O}\left(\frac{\overline{\tau}/\tau_{\text{eff}}}{\sqrt{m\overline{\tau}T}}\right) + \mathcal{O}\left(\frac{A\sigma^2}{\sqrt{m\overline{\tau}T}}\right) + \mathcal{O}\left(\frac{mB\sigma^2}{\overline{\tau}T}\right) + \mathcal{O}\left(\frac{mC\kappa^2}{\overline{\tau}T}\right)}_{\text{denoted by } \epsilon_{opt} \text{ in (10)}} \quad (8)$$

*where $\mathcal{O}$ swallows all constants (including $L$), and quantities $A, B, C$ are defined as follows:*

$$A = m\tau_{\text{eff}} \sum_{i=1}^m \frac{w_i^2 \|\boldsymbol{a}_i\|_2^2}{\|\boldsymbol{a}_i\|_1^2}, \ B = \sum_{i=1}^m w_i(\|\boldsymbol{a}_i\|_2^2 - a_{i,-1}^2), \ C = \max_i\{\|\boldsymbol{a}_i\|_1^2 - \|\boldsymbol{a}_i\|_1 a_{i,-1}\} \quad (9)$$

*where $a_{i,-1}$ is the last element in the vector $\boldsymbol{a}_i$.*

In the Appendix, we also provide another version of this theorem that explicitly contains the local learning rate $\eta$. Moreover, since the surrogate objective $\widetilde{F}(\boldsymbol{x})$ and the original objective $F(\boldsymbol{x})$ are just different linear combinations of the local functions, once the algorithm converges to a stationary point of $\widetilde{F}(\boldsymbol{x})$, one can also obtain some guarantees in terms of $F(\boldsymbol{x})$, as given by Theorem 2 below.

**Theorem 2** (**Convergence in Terms of the True Objective** $F(\boldsymbol{x})$). *Under the same conditions as Theorem 1, the minimal gradient norm of the true global objective function $F(\boldsymbol{x}) = \sum_{i=1}^{m} p_i F_i(\boldsymbol{x})$ will be bounded as follows:*

$$\min_{t \in [T]} \|\nabla F(\boldsymbol{x}^{(t,0)})\|^2 \leq \underbrace{2\left[\chi_{\boldsymbol{p}\|\boldsymbol{w}}^2(\beta^2 - 1) + 1\right]\epsilon_{opt}}_{\text{vanishing error term}} + \underbrace{2\chi_{\boldsymbol{p}\|\boldsymbol{w}}^2\kappa^2}_{\text{non-vanishing error due to obj. inconsistency}} \tag{10}$$

*where $\epsilon_{opt}$ denotes the vanishing optimization error given by (8) and $\chi_{\boldsymbol{p}\|\boldsymbol{w}}^2 = \sum_{i=1}^{m}(p_i - w_i)^2/w_i$ represents the chi-square divergence between vectors $\boldsymbol{p} = [p_1, \ldots, p_m]$ and $\boldsymbol{w} = [w_1, \ldots, w_m]$.*

**Discussion:** Theorems 1 and 2 describe the convergence behavior of a broad class of federated heterogeneous optimization algorithms. Observe that when all clients take the same number of local steps using the same local solver, we have $\boldsymbol{p} = \boldsymbol{w}$ such that $\chi^2 = 0$. Also, when all local functions are identical to each other, we have $\beta^2 = 1, \kappa^2 = 0$. Only in these two special cases, is there no objective inconsistency. For most other algorithms subsumed by the general update rule in (4), both $w_i$ and $\tau_{\text{eff}}$ are influenced by the choice of $\boldsymbol{a}_i$. When clients have different local progress (*i.e.*, different $\boldsymbol{a}_i$ vectors), previous algorithms will end up with a non-zero error floor $\chi^2\kappa^2$, which does not vanish to 0 even with sufficiently small learning rate. In Appendix, we further construct a lower bound and show that $\lim_{T \to \infty} \min_{t \in [T]} \|\nabla F(\boldsymbol{x}^{(t,0)})\|^2 = \Omega(\chi_{\boldsymbol{p}\|\boldsymbol{w}}^2\kappa^2)$, suggesting (10) is tight.

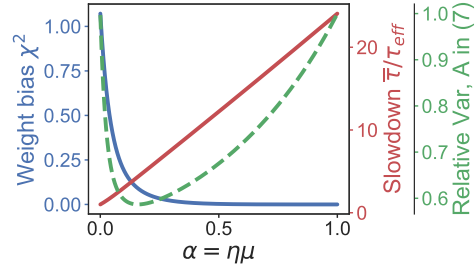

Figure 4: Illustration on how the parameter $\alpha = \eta\mu$ influences the convergence of FedProx. We set $m = 30, p_i = 1/m, \tau_i \sim \mathcal{N}(20, 20)$. '*Weight bias*' denotes the chi-square distance between $\boldsymbol{p}$ and $\boldsymbol{w}$. '*Slow-down*' and '*Relative Variance*' quantify how the first and the second terms in (8) change.

**Novel Insights Into the Convergence of FedProx and the Effect of** $\mu$. Recall that in FedProx $\boldsymbol{a}_i = [(1 - \alpha)^{\tau_i - 1}, \ldots, (1 - \alpha), 1]$, where $\alpha = \eta\mu$. Accordingly, substituting the effective steps and aggregated weight, given by (7), into (8) and (10), we get the convergence guarantee for FedProx. Again, it has objective inconsistency because $w_i \neq p_i$. As we increase $\alpha$, the weights $w_i$ come closer to $p_i$ and thus, the non-vanishing error $\chi^2\kappa^2$ in (10) decreases (see blue curve in Figure 4). However increasing $\alpha$ worsens the slowdown $\overline{\tau}/\tau_{\text{eff}}$, which appears in the first error term in (8) (see the red curve in Figure 4). In the extreme case when $\alpha = 1$, although FedProx achieves objective consistency, it has a significantly slower convergence because $\tau_{\text{eff}} = 1$ and the first term in (8) is $\overline{\tau}$ times larger than that with FedAvg (eq. to $\alpha = 0$).

Theorem 1 also reveals that, in FedProx, there should exist a best value of $\alpha$ that balances all terms in (8). In Appendix, we provide a corollary showing that $\alpha = \mathcal{O}(m^{\frac{1}{2}}/\overline{\tau}^{\frac{1}{2}}T^{\frac{1}{6}})$ optimizes the error bound (8) of FedProx and yields a convergence rate of $\mathcal{O}(1/\sqrt{m\overline{\tau}T} + 1/T^{\frac{2}{3}})$ on the surrogate objective. This can serve as a guideline on setting $\alpha$ in practice.

**Linear Speedup Analysis.** Another implication of Theorem 1 is that when the communication rounds $T$ is sufficiently large, then the convergence of the surrogate objective will be dominated by the first two terms in (8), which is $1/\sqrt{m\overline{\tau}T}$. This suggests that the algorithm only uses $T/\gamma$ total rounds when using $\gamma$ times more clients (*i.e.*, achieving linear speedup) to reach the same error level.

## 5   FedNova: Proposed Federated Normalized Averaging Algorithm

Theorems 1 and 2 suggest an extremely simple solution to overcome the problem of objective inconsistency. When we set $w_i = p_i$ in (4), then the second non-vanishing term $\chi_{\boldsymbol{p}\|\boldsymbol{w}}^2\kappa^2$ in (10) will

just become zero. This simple intuition yields the following new algorithm:

$$\texttt{FedNova} \quad \boldsymbol{x}^{(t+1,0)} - \boldsymbol{x}^{(t,0)} = -\tau_{\text{eff}}^{(t)} \sum_{i=1}^{m} p_i \cdot \eta \boldsymbol{d}_i^{(t)} \quad \text{where } \boldsymbol{d}_i^{(t)} = \boldsymbol{G}_i^{(t)} \boldsymbol{a}_i^{(t)} / \|\boldsymbol{a}_i^{(t)}\|_1 \quad (11)$$

The proposed algorithm is named *federated normalized averaging* (FedNova), because the locally normalized updates $\boldsymbol{d}_i$ are averaged/aggregated instead of the local changes $\Delta_i = -\eta \boldsymbol{G}_i \boldsymbol{a}_i$. When the local solver is vanilla SGD, then $\boldsymbol{a}_i = [1, 1, \dots, 1] \in \mathbb{R}^{\tau_i}$ and $\boldsymbol{d}_i^{(t)}$ is a simple average over current round's gradients. In order to be consistent with FedAvg whose update rule is (6), one can simply set $\tau_{\text{eff}}^{(t)} = \sum_{i=1}^{m} p_i \tau_i^{(t)}$. Then, in this case, the update rule of FedNova is equivalent to

$$\boldsymbol{x}^{(t+1,0)} - \boldsymbol{x}^{(t,0)} = \left( \sum_{i=1}^{m} p_i \tau_i^{(t)} \right) \sum_{i=1}^{m} p_i \frac{\Delta_i^{(t)}}{\tau_i^{(t)}}. \quad (12)$$

Comparing to previous algorithm $\boldsymbol{x}^{(t+1,0)} - \boldsymbol{x}^{(t,0)} = \sum_{i=1}^{m} p_i \Delta_i^{(t)}$, each local change in FedNova is re-scaled by $(\sum_{i=1}^{m} p_i \tau_i^{(t)}) / \tau_i^{(t)}$. This simple tweak in the aggregation weights eliminates inconsistency in the solution.

**Flexibility in Choosing Hyper-parameters and Local Solvers.** Besides vanilla SGD, the new formulation of FedNova naturally allows clients to choose various local solvers (*i.e.*, client-side optimizer). As discussed in Section 4.1, the local solver can also be GD/SGD with decayed local learning rate, GD/SGD with proximal updates, GD/SGD with local momentum, etc. Furthermore, the value of $\tau_{\text{eff}}$ is not necessarily to be controlled by the local solver as previous algorithms. For example, when using SGD with proximal updates, one can simply set $\tau_{\text{eff}} = \sum_{i=1}^{m} p_i \tau_i$ instead of its default value $\sum_{i=1}^{m} p_i [1 - (1 - \alpha)^{\tau_i}] / \alpha$. This can help alleviate the slowdown problem discussed in Section 4.2.

**Combination with Acceleration Techniques.** If clients are stateful and have additional communication bandwidth, they can use cross-client variance reduction techniques to further accelerate the training [21, 20, 39]. In this case, the local gradient at the $k$-th local step becomes $g_i(\boldsymbol{x}^{(t,k)}) + \sum_{i=1}^{m} p_i \boldsymbol{d}_i^{(t-1)} - \boldsymbol{d}_i^{(t-1)}$. Besides, on the server side, one can also implement server momentum or adaptive server optimizers [25, 42, 40], in which the aggregated normalized gradient $-\tau_{\text{eff}} \sum_{i=1}^{m} \eta p_i \boldsymbol{d}_i$ is used to update the server momentum buffer instead of directly updating the server model.

**Convergence Analysis.** The local solvers at clients do not necessarily need to be the same or fixed across rounds. In the following theorem, we obtain strong convergence guarantee for FedNova, even with *arbitrarily time-varying* local updates and client optimizers.

**Theorem 3** (**Convergence of** FedNova **to a Consistent Solution**). *Suppose that each client performs arbitrary number of local updates $\tau_i(t)$ using arbitrary gradient accumulation method $\boldsymbol{a}_i(t), t \in [T]$ per round. Under Assumptions 1 to 3, if local learning rate is set as $\eta = \sqrt{m^2/K}$, where $K = m \sum_{t=0}^{T-1} \tau_i(t)$ denotes the number of processed mini-batches across all clients after $T$ rounds, then FedNova converges to a stationary point of $F(\boldsymbol{x})$. The detailed bound is the same as the right hand side of (8), except that $\overline{\tau}, A, B, C$ are replaced by their average values over all rounds.*

Using the techniques developed in [12, 20, 13], Theorem 3 can be further generalized to incorporate client sampling schemes. We provide corresponding corollaries in the Appendix. Moreover, forcing all clients to perform $\tau = \min_i \tau_i$ local steps (let us call this algorithm FedAvg-min) can also ensure objective consistency. However, in each round, FedAvg-min will go over less data samples than FedNova ($mb\tau_{\min}$ versus $b \sum_{i=1}^{m} \tau_i$ where $b$ is the mini-batch size), resulting in worse performance. Another drawback of a fixed $\tau$ algorithm like FedAvg-min is that faster nodes would remain idle in each round while waiting for slower nodes. FedNova avoids such straggling delays by allowing nodes to make different numbers of local updates.

## 6 Experimental Results

**Experimental Setup.** We evaluate all algorithms on two setups with non-IID data partitioning: (1) *Logistic Regression on a Synthetic Federated Dataset*: The dataset Synthetic(1, 1) is originally constructed in [38]. The local dataset sizes $n_i, i \in [1, 30]$ follows a power law. (2) *DNN trained on a Non-IID partitioned CIFAR-10 dataset*: We train a VGG-11 [52] network on the CIFAR-10 dataset [53], which is partitioned across 16 clients using a Dirichlet distribution $\text{Dir}_{16}(0.1)$,

as done in [54]. The original CIFAR-10 test set (without partitioning) is used to evaluate the generalization performance of the trained global model. The local learning rate $\eta$ is decayed by a constant factor after finishing 50% and 75% of the communication rounds. The initial value of $\eta$ is tuned separately for FedAvg with different local solvers. When using the same solver, FedNova uses the same $\eta$ as FedAvg to guarantee a fair comparison. On CIFAR-10, we run each experiment with 3 random seeds and report the average and standard deviation. More details are in Appendix[2].

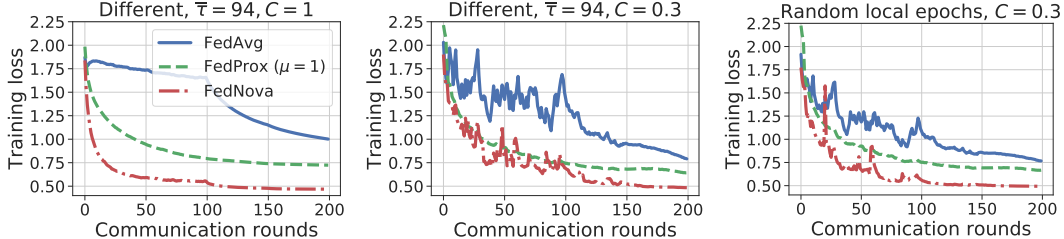

Figure 5: Results on the synthetic dataset under three different settings. In FedProx, we set $\mu = 1$, the best value reported in [38]. **Left**: All clients perform $E_i = 5$ local epochs; **Middle**: Only $C = 0.3$ fraction of clients are randomly selected per round to perform $E_i = 5$ local epochs; **Right**: Only $C = 0.3$ fraction of clients are randomly selected per round to perform random and time-varying local epochs $E_i(t) \sim \mathcal{U}(1, 5)$.

**Synthetic Dataset Simulations.** In Figure 5, we observe that by simply changing $w_i$ to $p_i$, FedNova not only converges faster than FedAvg but also achieves consistently the best performance under three different settings. Note that the only difference between FedNova and FedAvg is the aggregated weights when averaging the normalized gradients.

**Non-IID CIFAR-10 Experiments.** In Table 1 we compare the performance of FedNova and FedAvg on non-IID CIFAR-10 with various client optimizers run for 100 communication rounds. When the client optimizer is SGD or SGD with momentum, simply changing the weights yields a 6-9% improvement on the test

Table 1: Results comparing FedAvg and FedNova with various client optimizers (*i.e.*, local solvers) trained on non-IID CIFAR-10 dataset. FedProx and SCAFFOLD correspond to FedAvg with proximal SGD updates and cross-client variance-reduction (VR), respectively.

| Local Epochs | Client Opt. | Test Accuracy % | |
|---|---|---|---|
| | | FedAvg | FedNova |
| $E_i = 2$ $(16 \leq \tau_i \leq 408)$ | Vanilla | $60.68\pm1.05$ | $\mathbf{66.31}\pm0.86$ |
| | Momentum | $65.26\pm2.42$ | $\mathbf{73.32}\pm0.29$ |
| | Proximal [38] | $60.44\pm1.21$ | $\mathbf{69.92}\pm0.34$ |
| $E_i^{(t)} \sim \mathcal{U}(2,5)$ $(16 \leq \tau_i^{(t)} \leq 1020)$ | Vanilla | $64.22\pm1.06$ | $\mathbf{73.22}\pm0.32$ |
| | Momentum | $70.44\pm2.99$ | $\mathbf{77.07}\pm0.12$ |
| | Proximal [38] | $63.74\pm1.44$ | $\mathbf{73.41}\pm0.45$ |
| | VR [20] | $74.72\pm0.34$ | $74.72\pm0.19$ |
| | Momen.+VR | Not Defined | $\mathbf{79.19}\pm0.17$ |

accuracy; When the client optimizer is proximal SGD, FedAvg is equivalent to FedProx. We manually tune the value of $\mu$ from $\{0.0005, 0.001, 0.005, 0.01\}$. By setting $\tau_{\text{eff}} = \sum_{i=1}^{m} p_i \tau_i$ and correcting the weights $w_i = p_i$ while keeping $\boldsymbol{a}_i$ same as FedProx, FedNova-Prox achieves about 10% higher test accuracy than FedProx. When using variance-reduction methods such as SCAFFOLD (that requires doubled communication), FedNova-based method preserves the same test accuracy. Furthermore, combining local momentum and variance-reduction in FedNova achieves the highest test accuracy among all other solvers. This kind of combination is non-trivial and has not appeared yet in the literature. We provide its pseudo-code in the Appendix.

**Effectiveness of Local Momentum.** From Table 1, it is worth noting that using momentum SGD as the local solver is an effective way to improve the performance. It generally achieves 3-7% higher test accuracy than vanilla SGD. This local momentum scheme can be further combined with server momentum [25, 42, 40]. When $E_i(t) \sim \mathcal{U}(2, 5)$, the hybrid momentum scheme achieves test accuracy $81.15 \pm 0.38\%$ As a reference, using server momentum alone achieves $77.49 \pm 0.25\%$.

## Broader Impact

The future of machine learning lies in moving both data collection as well as model training to the edge. This nascent research field called federated learning considers a large number of resource-constrained devices such as cellphones or IoT sensors that collect training data from their environment. Due to limited communication capabilities as well as privacy concerns, these data cannot be directly sent over to the cloud. Instead, the nodes locally perform a few iterations of training and only send the resulting model to the cloud. In this paper, we develop a federated training algorithm that is system-aware (robust and adaptable to communication and computation variabilities by allowing heterogeneous local progress) and data-aware (can handle skews in the size and distribution of local training data by correcting model aggregation scheme). This research has the potential to democratize machine learning by transcending the current centralized machine learning framework. It will enable lightweight mobile devices to cooperatively train a common machine learning model while maintaining control of their training data.

## Acknowledgments and Disclosure of Funding

This research was generously supported in part by NSF grants CCF-1850029, the 2018 IBM Faculty Research Award, and the Qualcomm Innovation fellowship (Jianyu Wang). We thank Anit Kumar Sahu, Tian Li, Zachary Charles, Zachary Garrett, and Virginia Smith for helpful discussions.

## Footnotes

[1] In the case of client sampling, the update rule of `FedAvg` (2) should hold in expectation in order to guarantee convergence [12, 13, 38, 40]. One can achieve this by either (i) sampling $q$ clients with replacement with respect to probability $p_i$, and then averaging the cumulative local changes with equal weights, or (ii) sampling $q$ clients without replacement uniformly at random, and then weighted averaging local changes, where the weight of client $i$ is re-scaled to $p_i m/q$. Our convergence analysis can be easily extended to these two cases.

[2]Our code is available at: https://github.com/JYWa/FedNova.

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
