[Supplementary Material]

## A  Proof of Lemma 1: Objective Inconsistency in Quadratic Model

**Formulation.**  Consider a simple setting where each local objective function is strongly convex and defined as follows:

$$F_i(\boldsymbol{x}) = \frac{1}{2}\boldsymbol{x}^\top \boldsymbol{H}_i \boldsymbol{x} - \boldsymbol{e}_i^\top \boldsymbol{x} + \frac{1}{2}\boldsymbol{e}_i^\top \boldsymbol{H}_i^{-1}\boldsymbol{e}_i \geq 0 \tag{13}$$

where $\boldsymbol{H}_i \in \mathbb{R}^{d \times d}$ is an invertible matrix and $\boldsymbol{e}_i \in \mathbb{R}^d$ is an arbitrary vector. It is easy to show that the optimum of the $i$-th local function is $\boldsymbol{x}_i^* = \boldsymbol{H}_i^{-1}\boldsymbol{e}_i$. Without loss of generality, we assume the global objective function to be a weighted average across all local functions, that is:

$$F(\boldsymbol{x}) = \sum_{i=1}^{m} p_i F_i(\boldsymbol{x}) = \frac{1}{2}\boldsymbol{x}^\top \overline{\boldsymbol{H}}\boldsymbol{x} - \overline{\boldsymbol{e}}^\top \boldsymbol{x} + \frac{1}{2}\sum_{i=1}^{m} p_i \boldsymbol{e}_i^\top \boldsymbol{h}_i^{-1}\boldsymbol{e}_i \tag{14}$$

where $\overline{\boldsymbol{H}} = \sum_{i=1}^{m} p_i \boldsymbol{H}_i$ and $\overline{\boldsymbol{e}} = \sum_{i=1}^{m} p_i \boldsymbol{e}_i$. As a result, the global minimum is $\boldsymbol{x}^* = \overline{\boldsymbol{H}}^{-1}\overline{\boldsymbol{e}}$. Now, let us study whether previous federated optimization algorithms can converge to this global minimum.

**Local Update Rule.**  The local update rule of FedProx for the $i$-th device can be written as follows:

$$\boldsymbol{x}_i^{(t,k+1)} = \boldsymbol{x}_i^{(t,k)} - \eta \left[ \boldsymbol{H}_i \boldsymbol{x}_i^{(t,k)} - \boldsymbol{e}_i + \mu(\boldsymbol{x}_i^{(t,k)} - \boldsymbol{x}^{(t,0)}) \right] \tag{15}$$

$$= (\boldsymbol{I} - \eta\mu\boldsymbol{I} - \eta\boldsymbol{H}_i)\boldsymbol{x}_i^{(t,k)} + \eta\boldsymbol{e}_i + \eta\mu\boldsymbol{x}^{(t,0)} \tag{16}$$

where $\boldsymbol{x}_i^{(t,k)}$ denotes the local model parameters at the $k$-th local iteration after $t$ communication rounds, $\eta$ denotes the local learning rate and $\mu$ is a tunable hyper-parameter in FedProx. When $\mu = 0$, the algorithm will reduce to FedAvg. We omit the device index in $\boldsymbol{x}^{(t,0)}$, since it is synchronized and the same across all devices.

After minor arranging (16), we obtain

$$\boldsymbol{x}_i^{(t,k+1)} - \boldsymbol{c}_i^{(t)} = (\boldsymbol{I} - \eta\mu\boldsymbol{I} - \eta\boldsymbol{H}_i) \left( \boldsymbol{x}_i^{(t,k)} - \boldsymbol{c}_i^{(t)} \right). \tag{17}$$

where $\boldsymbol{c}_i^{(t)} = (\boldsymbol{H}_i + \mu\boldsymbol{I})^{-1} \left( \boldsymbol{e}_i + \mu\boldsymbol{x}^{(t,0)} \right)$. Then, after performing $\tau_i$ steps of local updates, the local model becomes

$$\boldsymbol{x}_i^{(t,\tau_i)} = (\boldsymbol{I} - \eta\mu\boldsymbol{I} - \eta\boldsymbol{H}_i)^{\tau_i} \left( \boldsymbol{x}^{(t,0)} - \boldsymbol{c}_i^{(t)} \right) + \boldsymbol{c}_i^{(t)}, \tag{18}$$

$$\boldsymbol{x}_i^{(t,\tau_i)} - \boldsymbol{x}^{(t,0)} = (\boldsymbol{I} - \eta\mu\boldsymbol{I} - \eta\boldsymbol{H}_i)^{\tau_i} \left( \boldsymbol{x}^{(t,0)} - \boldsymbol{c}_i^{(t)} \right) + \boldsymbol{c}_i^{(t)} - \boldsymbol{x}^{(t,0)} \tag{19}$$

$$= [(\boldsymbol{I} - \eta\mu\boldsymbol{I} - \eta\boldsymbol{H}_i)^{\tau_i} - \boldsymbol{I}] \left( \boldsymbol{x}^{(t,0)} - \boldsymbol{c}_i^{(t)} \right) \tag{20}$$

$$= [\boldsymbol{I} - (\boldsymbol{I} - \eta\mu\boldsymbol{I} - \eta\boldsymbol{H}_i)^{\tau_i}] (\boldsymbol{H}_i + \mu\boldsymbol{I})^{-1} \left( \boldsymbol{e}_i - \boldsymbol{H}_i \boldsymbol{x}^{(t,0)} \right). \tag{21}$$

For the ease of writing, we define $\boldsymbol{K}_i(\eta, \mu) = [\boldsymbol{I} - (\boldsymbol{I} - \eta\mu\boldsymbol{I} - \eta\boldsymbol{H}_i)^{\tau_i}] (\boldsymbol{H}_i + \mu\boldsymbol{I})^{-1}$.

**Server Aggregation.**  For simplicity, we only consider the case when all devices participate in the each round. In FedProx, the server averages all local models according to the sample size:

$$\boldsymbol{x}^{(t+1,0)} - \boldsymbol{x}^{(t,0)} = \sum_{i=1}^{m} p_i \left( \boldsymbol{x}_i^{(t,\tau_i)} - \boldsymbol{x}^{(t,0)} \right) \tag{22}$$

$$= \sum_{i=1}^{m} p_i \boldsymbol{K}_i(\eta, \mu) \left( \boldsymbol{e}_i - \boldsymbol{H}_i \boldsymbol{x}^{(t,0)} \right). \tag{23}$$

Accordingly, we get the following update rule for the central model:

$$\boldsymbol{x}^{(t+1,0)} = \left[ \boldsymbol{I} - \sum_{i=1}^{m} p_i \boldsymbol{K}_i(\eta, \mu) \boldsymbol{H}_i \right] \boldsymbol{x}^{(t,0)} + \sum_{i=1}^{m} p_i \boldsymbol{K}_i(\eta, \mu) \boldsymbol{e}_i. \tag{24}$$

It is equivalent to

$$\boldsymbol{x}^{(t+1,0)} - \widetilde{\boldsymbol{x}} = \left[ \boldsymbol{I} - \sum_{i=1}^{m} p_i \boldsymbol{K}_i(\eta, \mu) \boldsymbol{H}_i \right] \left[ \boldsymbol{x}^{(t,0)} - \widetilde{\boldsymbol{x}} \right]. \tag{25}$$

where

$$\widetilde{\boldsymbol{x}} = \left( \sum_{i=1}^{m} p_i \boldsymbol{K}_i(\eta, \mu) \boldsymbol{H}_i \right)^{-1} \left( \sum_{i=1}^{m} p_i \boldsymbol{K}_i(\eta, \mu) \boldsymbol{e}_i \right). \tag{26}$$

After $T$ communication rounds, one can get

$$\boldsymbol{x}^{(T,0)} = \left[ \boldsymbol{I} - \sum_{i=1}^{m} p_i \boldsymbol{K}_i(\eta, \mu) \boldsymbol{H}_i \right]^{T} \left[ \boldsymbol{x}^{(t,0)} - \widetilde{\boldsymbol{x}} \right] + \widetilde{\boldsymbol{x}}. \tag{27}$$

Accordingly, when $\left\| \boldsymbol{I} - \sum_{i=1}^{m} p_i \boldsymbol{K}_i(\eta, \mu) \boldsymbol{H}_i \right\|_2 < 1$, the iterates will converge to

$$\lim_{T \to \infty} \boldsymbol{x}^{(T,0)} = \widetilde{\boldsymbol{x}} = \left( \sum_{i=1}^{m} p_i \boldsymbol{K}_i(\eta, \mu) \boldsymbol{H}_i \right)^{-1} \left( \sum_{i=1}^{m} p_i \boldsymbol{K}_i(\eta, \mu) \boldsymbol{e}_i \right). \tag{28}$$

Recall that $\boldsymbol{K}_i(\eta, \mu) = \left[ \boldsymbol{I} - (\boldsymbol{I} - \eta\mu\boldsymbol{I} - \eta\boldsymbol{H}_i)^{\tau_i} \right] (\boldsymbol{H}_i + \mu\boldsymbol{I})^{-1}$.

**Concrete Example in Lemma 1.** Now let us focus on a concrete example where $p_1 = p_2 = \cdots = p_m = 1/m$, $\boldsymbol{H}_1 = \boldsymbol{H}_2 = \cdots = \boldsymbol{H}_m = \boldsymbol{I}$ and $\mu = 0$. Then, in this case, $\boldsymbol{K}_i = 1 - (1 - \eta)^{\tau_i}$. As a result, we have

$$\lim_{T \to \infty} \boldsymbol{x}^{(T,0)} = \frac{\sum_{i=1}^{m} [1 - (1 - \eta)^{\tau_i}] \boldsymbol{e}_i}{\sum_{i=1}^{m} [1 - (1 - \eta)^{\tau_i}]}. \tag{29}$$

Furthermore, when the learning rate is sufficiently small (*e.g.*, can be achieved by gradually decaying the learning rate), according to L'Hospital's rule, we obtain

$$\lim_{\eta \to 0} \lim_{T \to \infty} \boldsymbol{x}^{(T,0)} = \frac{\sum_{i=1}^{m} \tau_i \boldsymbol{e}_i}{\sum_{i=1}^{m} \tau_i}. \tag{30}$$

Here, we complete the proof of Lemma 1.

# B  Detailed Derivations for Various Local Solvers

In this section, we will derive the specific expression of the vector $\boldsymbol{a}_i$ when using different local solvers. Recall that the local change at client $i$ is $\Delta_i^{(t)} = -\eta \boldsymbol{G}_i^{(t)} \boldsymbol{a}_i$ where $\boldsymbol{G}_i^{(t)}$ stacks all stochastic gradients in the current round and $\boldsymbol{a}$ is a non-negative vector.

## B.1  SGD with Proximal Updates

In this case, we can write the update rule of local models as follows:

$$\boldsymbol{x}_i^{(t,\tau_i)} = \boldsymbol{x}_i^{(t,\tau_i-1)} - \eta \left[ g_i(\boldsymbol{x}_i^{(t,\tau_i-1)}) + \mu \left( \boldsymbol{x}_i^{(t,\tau_i-1)} - \boldsymbol{x}^{(t,0)} \right) \right]. \tag{31}$$

Subtracting $\boldsymbol{x}_i^{(t,0)}$ on both sides, we obtain

$$\boldsymbol{x}_i^{(t,\tau_i)} - \boldsymbol{x}^{(t,0)} = \boldsymbol{x}_i^{(t,\tau_i-1)} - \boldsymbol{x}^{(t,0)} - \eta \left[ g_i(\boldsymbol{x}_i^{(t,\tau_i-1)}) + \mu \left( \boldsymbol{x}_i^{(t,\tau_i-1)} - \boldsymbol{x}^{(t,0)} \right) \right] \tag{32}$$

$$= (1 - \eta\mu) \left( \boldsymbol{x}_i^{(t,\tau_i-1)} - \boldsymbol{x}^{(t,0)} \right) - \eta g_i(\boldsymbol{x}_i^{(t,\tau_i-1)}). \tag{33}$$

Repeating the above procedure, it follows that

$$\Delta_i^{(t)} = \boldsymbol{x}_i^{(t,\tau_i)} - \boldsymbol{x}^{(t,0)} = -\eta \sum_{k=0}^{\tau_i-1} (1 - \eta\mu)^{\tau_i-1-k} g_i(\boldsymbol{x}_i^{(t,k)}). \tag{34}$$

According to the definition, we have $\boldsymbol{a}_i = [(1 - \alpha)^{\tau_i-1}, (1 - \alpha)^{\tau_i-2}, \ldots, (1 - \alpha), 1]$ where $\alpha = \eta\mu$.

## B.2 SGD with Local Momentum

Let us firstly write down the update rule of the local models. Suppose that $\rho$ denotes the local momentum factor and $\boldsymbol{u}_i$ is the local momentum buffer at client $i$. Then, the update rule of local momentum SGD is:

$$\boldsymbol{u}_i^{(t,\tau_i)} = \rho \boldsymbol{u}_i^{(t,\tau_i-1)} + g_i(\boldsymbol{x}_i^{(t,\tau_i-1)}), \tag{35}$$

$$\boldsymbol{x}_i^{(t,\tau_i)} = \boldsymbol{x}_i^{(t,\tau_i-1)} - \eta \boldsymbol{u}_i^{(t,\tau_i)}. \tag{36}$$

One can expand the expression of local momentum buffer as follows:

$$\boldsymbol{u}_i^{(t,\tau_i)} = \rho \boldsymbol{u}_i^{(t,\tau_i-1)} + g_i(\boldsymbol{x}_i^{(t,\tau_i-1)}) \tag{37}$$

$$= \rho^2 \boldsymbol{u}_i^{(t,\tau_i-2)} + \rho g_i(\boldsymbol{x}_i^{(t,\tau_i-2)}) + g_i(\boldsymbol{x}_i^{(t,\tau_i-1)}) \tag{38}$$

$$= \sum_{k=0}^{\tau_i-1} \rho^{\tau_i-1-k} g_i(\boldsymbol{x}_i^{(t,k)}) \tag{39}$$

where the last equation comes from the fact $\boldsymbol{u}_i^{(t,0)} = 0$. Substituting (39) into (36), we have

$$\boldsymbol{x}_i^{(t,\tau_i)} = \boldsymbol{x}_i^{(t,\tau_i-1)} - \eta \sum_{k=0}^{\tau_i-1} \rho^{\tau_i-1-k} g_i(\boldsymbol{x}_i^{(t,k)}) \tag{40}$$

$$= \boldsymbol{x}_i^{(t,\tau_i-2)} - \eta \sum_{k=0}^{\tau_i-2} \rho^{\tau_i-2-k} g_i(\boldsymbol{x}_i^{(t,k)}) - \eta \sum_{k=0}^{\tau_i-1} \rho^{\tau_i-1-k} g_i(\boldsymbol{x}_i^{(t,k)}). \tag{41}$$

Repeating the above procedure, it follows that

$$\boldsymbol{x}_i^{(t,\tau_i)} - \boldsymbol{x}^{(t,0)} = -\eta \sum_{s=0}^{\tau_i-1} \sum_{k=0}^{s} \rho^{s-k} g_i(\boldsymbol{x}_i^{(t,k)}) \tag{42}$$

Then, the coefficient of $g_i(\boldsymbol{x}_i^{(t,k)})$ is

$$\sum_{s\geq k}^{\tau_i-1} \rho^{s-k} = 1 + \rho + \rho^2 + \cdots + \rho^{\tau_i-1-k} = \frac{1-\rho^{\tau_i-k}}{1-\rho}. \tag{43}$$

That is, $\boldsymbol{a}_i = [1-\rho^{\tau_i}, 1-\rho^{\tau_i-1}, \ldots, 1-\rho]/(1-\rho)$. In this case, the $\ell_1$ norm of $\boldsymbol{a}_i$ is

$$\|\boldsymbol{a}_i\|_1 = \frac{1}{1-\rho} \sum_{k=0}^{\tau_i-1} \left(1-\rho^{\tau_i-k}\right) = \frac{1}{1-\rho} \left(\tau_i - \sum_{k=0}^{\tau_i-1} \rho^{\tau_i-k}\right) \tag{44}$$

$$= \frac{1}{1-\rho} \left[\tau_i - \frac{\rho(1-\rho^{\tau_i})}{1-\rho}\right]. \tag{45}$$

# C  Proof of Theorem 1: Convergence of Surrogate Objective

## C.1  Preliminaries

For the ease of writing, let us define a surrogate objective function $\widetilde{F}(\boldsymbol{x}) = \sum_{i=1}^{m} w_i F_i(\boldsymbol{x})$, where $\sum_{i=1}^{m} w_i = 1$, and define the following auxiliary variables

$$\text{Normalized Stochastic Gradient:} \quad \boldsymbol{d}_i^{(t)} = \frac{1}{a_i} \sum_{k=0}^{\tau_i - 1} a_i^{(k)} g_i(\boldsymbol{x}_i^{(t,k)}), \tag{46}$$

$$\text{Normalized Gradient:} \quad \boldsymbol{h}_i^{(t)} = \frac{1}{a_i} \sum_{k=0}^{\tau_i - 1} a_i^{(k)} \nabla F_i(\boldsymbol{x}_i^{(t,k)}) \tag{47}$$

where $a_i^{(k)} \geq 0$ is an arbitrary scalar, $\boldsymbol{a}_i = [a_i^{(0)}, \ldots, a_{i,-1}]^\top$, and $a_i = \|\boldsymbol{a}_i\|_1$. Besides, one can show that $\mathbb{E}[\boldsymbol{d}_i^{(t)} - \boldsymbol{h}_i^{(t)}] = 0$. In addition, since workers are independent to each other, we have $\mathbb{E}\left\langle \boldsymbol{d}_i^{(t)} - \boldsymbol{h}_i^{(t)}, \boldsymbol{d}_j^{(t)} - \boldsymbol{h}_j^{(t)} \right\rangle = 0, \forall i \neq j$. Recall that the update rule of the global model can be written as follows:

$$\boldsymbol{x}^{(t+1,0)} - \boldsymbol{x}^{(t,0)} = -\tau_{\text{eff}}\eta \sum_{i=1}^{m} w_i \boldsymbol{d}_i^{(t)}. \tag{48}$$

According to the Lipschitz-smooth assumption, it follows that

$$\mathbb{E}\left[\widetilde{F}(\boldsymbol{x}^{(t+1,0)})\right] - \widetilde{F}(\boldsymbol{x}^{(t,0)})$$

$$\leq -\tau_{\text{eff}}\eta \underbrace{\mathbb{E}\left[\left\langle \nabla \widetilde{F}(\boldsymbol{x}^{(t,0)}), \sum_{i=1}^{m} w_i \boldsymbol{d}_i^{(t)} \right\rangle\right]}_{T_1} + \frac{\tau_{\text{eff}}^2 \eta^2 L}{2} \underbrace{\mathbb{E}\left[\left\|\sum_{i=1}^{m} w_i \boldsymbol{d}_i^{(t)}\right\|^2\right]}_{T_2} \tag{49}$$

where the expectation is taken over mini-batches $\xi_i^{(t,k)}, \forall i \in \{1, 2, \ldots, m\}, k \in \{0, 1, \ldots, \tau_i - 1\}$. Before diving into the detailed bounds for $T_1$ and $T_2$, we would like to firstly introduce several useful lemmas.

**Lemma 2.** *Suppose $\{A_k\}_{k=1}^{T}$ is a sequence of random matrices and $\mathbb{E}[A_k | A_{k-1}, A_{k-2}, \ldots, A_1] = \boldsymbol{0}, \forall k$. Then,*

$$\mathbb{E}\left[\left\|\sum_{k=1}^{T} A_k\right\|_F^2\right] = \sum_{k=1}^{T} \mathbb{E}\left[\|A_k\|_F^2\right]. \tag{50}$$

*Proof.*

$$\mathbb{E}\left[\left\|\sum_{k=1}^{T} A_k\right\|_F^2\right] = \sum_{k=1}^{T} \mathbb{E}\left[\|A_k\|_F^2\right] + \sum_{i=1}^{T}\sum_{j=1, j\neq i}^{T} \mathbb{E}\left[\text{Tr}\{A_i^\top A_j\}\right] \tag{51}$$

$$= \sum_{k=1}^{T} \mathbb{E}\left[\|A_k\|_F^2\right] + \sum_{i=1}^{T}\sum_{j=1, j\neq i}^{T} \text{Tr}\{\mathbb{E}\left[A_i^\top A_j\right]\} \tag{52}$$

Assume $i < j$. Then, using the law of total expectation,

$$\mathbb{E}\left[A_i^\top A_j\right] = \mathbb{E}\left[A_i^\top \mathbb{E}\left[A_j | A_i, \ldots, A_1\right]\right] = \boldsymbol{0}. \tag{53}$$

$\square$

## C.2 Bounding First term in (49)

For the first term on the right hand side (RHS) in (49), we have

$$T_1 = \mathbb{E}\left[\left\langle \nabla \widetilde{F}(\boldsymbol{x}^{(t,0)}), \sum_{i=1}^m w_i \left(\boldsymbol{d}_i^{(t)} - \boldsymbol{h}_i^{(t)}\right)\right\rangle\right] + \mathbb{E}\left[\left\langle \nabla \widetilde{F}(\boldsymbol{x}^{(t,0)}), \sum_{i=1}^m w_i \boldsymbol{h}_i^{(t)}\right\rangle\right] \tag{54}$$

$$= \mathbb{E}\left[\left\langle \nabla \widetilde{F}(\boldsymbol{x}^{(t,0)}), \sum_{i=1}^m w_i \boldsymbol{h}_i^{(t)}\right\rangle\right] \tag{55}$$

$$= \frac{1}{2}\left\|\nabla \widetilde{F}(\boldsymbol{x}^{(t)})\right\|^2 + \frac{1}{2}\mathbb{E}\left[\left\|\sum_{i=1}^m w_i \boldsymbol{h}_i^{(t)}\right\|^2\right] - \frac{1}{2}\mathbb{E}\left[\left\|\nabla \widetilde{F}(\boldsymbol{x}^{(t,0)}) - \sum_{i=1}^m w_i \boldsymbol{h}_i^{(t)}\right\|^2\right] \tag{56}$$

where the last equation uses the fact: $2\langle a, b\rangle = \|a\|^2 + \|b\|^2 - \|a-b\|^2$.

## C.3 Bounding Second term in (49)

For the second term on the right hand side (RHS) in (49), we have

$$T_2 = \mathbb{E}\left[\left\|\sum_{i=1}^m w_i \left(\boldsymbol{d}_i^{(t)} - \boldsymbol{h}_i^{(t)}\right) + \sum_{i=1}^m w_i \boldsymbol{h}_i^{(t)}\right\|^2\right] \tag{57}$$

$$\leq 2\mathbb{E}\left[\left\|\sum_{i=1}^m w_i \left(\boldsymbol{d}_i^{(t)} - \boldsymbol{h}_i^{(t)}\right)\right\|^2\right] + 2\mathbb{E}\left[\left\|\sum_{i=1}^m w_i \boldsymbol{h}_i^{(t)}\right\|^2\right] \tag{58}$$

$$= 2\sum_{i=1}^m w_i^2 \mathbb{E}\left[\left\|\boldsymbol{d}_i^{(t)} - \boldsymbol{h}_i^{(t)}\right\|^2\right] + 2\mathbb{E}\left[\left\|\sum_{i=1}^m w_i \boldsymbol{h}_i^{(t)}\right\|^2\right] \tag{59}$$

where (58) follows the fact: $\|a+b\|^2 \leq 2\|a\|^2 + 2\|b\|^2$ and (59) uses the special property of $\boldsymbol{d}_i^{(t)}, \boldsymbol{h}_i^{(t)}$, that is, $\mathbb{E}\left\langle \boldsymbol{d}_i^{(t)} - \boldsymbol{h}_i^{(t)}, \boldsymbol{d}_j^{(t)} - \boldsymbol{h}_j^{(t)}\right\rangle = 0, \forall i \neq j$. Then, let us expand the expression of $\boldsymbol{d}_i^{(t)}$ and $\boldsymbol{h}_i^{(t)}$, we obtain that

$$T_2 \leq \sum_{i=1}^m \frac{2w_i^2}{a_i^2} \sum_{k=0}^{\tau_i - 1} [a_i^{(k)}]^2 \mathbb{E}\left[\left\|g_i(\boldsymbol{x}_i^{(t,k)}) - \nabla F_i(\boldsymbol{x}_i^{(t,k)})\right\|^2\right] + 2\mathbb{E}\left[\left\|\sum_{i=1}^m w_i \boldsymbol{h}_i^{(t)}\right\|^2\right] \tag{60}$$

$$\leq 2\sigma^2 \sum_{i=1}^m \frac{w_i^2 \|\boldsymbol{a}_i\|^2}{\|\boldsymbol{a}_i\|_1^2} + 2\mathbb{E}\left[\left\|\sum_{i=1}^m w_i \boldsymbol{h}_i^{(t)}\right\|^2\right] \tag{61}$$

where (60) is derived using Lemma 2, (61) follows Assumption 2.

## C.4 Intermediate Result

Plugging (56) and (61) back into (49), we have

$$\mathbb{E}\left[\widetilde{F}(\boldsymbol{x}^{(t+1,0)})\right] - \widetilde{F}(\boldsymbol{x}^{(t,0)}) \leq -\frac{\tau_{\text{eff}}\eta}{2}\left\|\nabla \widetilde{F}(\boldsymbol{x}^{(t,0)})\right\|^2 - \frac{\tau_{\text{eff}}\eta}{2}\left(1 - 2\tau_{\text{eff}}\eta L\right)\mathbb{E}\left[\left\|\sum_{i=1}^m w_i \boldsymbol{h}_i^{(t)}\right\|^2\right]$$

$$+ \tau_{\text{eff}}^2 \eta^2 L\sigma^2 \sum_{i=1}^m \frac{w_i^2 \|\boldsymbol{a}_i\|_2^2}{\|\boldsymbol{a}_i\|_1^2} + \frac{\tau_{\text{eff}}\eta}{2}\mathbb{E}\left[\left\|\nabla \widetilde{F}(\boldsymbol{x}^{(t,0)}) - \sum_{i=1}^m w_i \boldsymbol{h}_i^{(t)}\right\|^2\right] \tag{62}$$

When $\tau_{\text{eff}}\eta L \leq 1/2$, it follows that

$$\frac{\mathbb{E}\left[\widetilde{F}(\boldsymbol{x}^{(t+1,0)})\right] - \widetilde{F}(\boldsymbol{x}^{(t,0)})}{\eta\tau_{\text{eff}}} \leq -\frac{1}{2}\left\|\nabla\widetilde{F}(\boldsymbol{x}^{(t,0)})\right\|^2 + \tau_{\text{eff}}\eta L\sigma^2 \sum_{i=1}^{m}\frac{w_i^2\|\boldsymbol{a}_i\|_2^2}{\|\boldsymbol{a}_i\|_1^2}$$

$$+\frac{1}{2}\mathbb{E}\left[\left\|\nabla\widetilde{F}(\boldsymbol{x}^{(t,0)}) - \sum_{i=1}^{m}w_i\boldsymbol{h}_i^{(t)}\right\|^2\right] \tag{63}$$

$$\leq -\frac{1}{2}\left\|\nabla\widetilde{F}(\boldsymbol{x}^{(t,0)})\right\|^2 + \tau_{\text{eff}}\eta L\sigma^2 \sum_{i=1}^{m}\frac{w_i^2\|\boldsymbol{a}_i\|_2^2}{\|\boldsymbol{a}_i\|_1^2}$$

$$+\frac{1}{2}\sum_{i=1}^{m}w_i\mathbb{E}\left[\left\|\nabla F_i(\boldsymbol{x}^{(t,0)}) - \boldsymbol{h}_i^{(t)}\right\|^2\right] \tag{64}$$

where the last inequality uses the fact $\widetilde{F}(\boldsymbol{x}) = \sum_{i=1}^{m}w_iF_i(\boldsymbol{x})$ and Jensen's Inequality: $\|\sum_{i=1}^{m}w_iz_i\|^2 \leq \sum_{i=1}^{m}w_i\|z_i\|^2$. Next, we will focus on bounding the last term in (64).

## C.5 Bounding the Difference Between Server Gradient and Normalized Gradient

Recall the definition of $\boldsymbol{h}_i^{(t)}$, one can derive that

$$\mathbb{E}\left[\left\|\nabla F_i(\boldsymbol{x}^{(t,0)}) - \boldsymbol{h}_i^{(t)}\right\|^2\right] = \mathbb{E}\left[\left\|\nabla F_i(\boldsymbol{x}^{(t,0)}) - \frac{1}{a_i}\sum_{k=0}^{\tau_i-1}a_i^{(k)}\nabla F_i(\boldsymbol{x}_i^{(t,k)})\right\|^2\right] \tag{65}$$

$$=\mathbb{E}\left[\left\|\frac{1}{a_i}\sum_{k=0}^{\tau_i-1}a_i^{(k)}\left(\nabla F_i(\boldsymbol{x}^{(t,0)}) - \nabla F_i(\boldsymbol{x}_i^{(t,k)})\right)\right\|^2\right] \tag{66}$$

$$\leq\frac{1}{a_i}\sum_{k=0}^{\tau_i-1}\left\{a_i^{(k)}\mathbb{E}\left[\left\|\nabla F_i(\boldsymbol{x}^{(t,0)}) - \nabla F_i(\boldsymbol{x}_i^{(t,k)})\right\|^2\right]\right\} \tag{67}$$

$$\leq\frac{L^2}{a_i}\sum_{k=0}^{\tau_i-1}\left\{a_i^{(k)}\mathbb{E}\left[\left\|\boldsymbol{x}^{(t,0)} - \boldsymbol{x}_i^{(t,k)}\right\|^2\right]\right\} \tag{68}$$

where (67) uses Jensen's Inequality again: $\|\sum_{i=1}^{m}w_iz_i\|^2 \leq \sum_{i=1}^{m}w_i\|z_i\|^2$, and (68) follows Assumption 1. Now, we turn to bounding the difference between the server model $\boldsymbol{x}^{(t,0)}$ and the local model $\boldsymbol{x}_i^{(t,k)}$. Plugging into the local update rule and using the fact $\|a+b\|^2 \leq 2\|a\|^2 + 2\|b\|^2$,

$$\mathbb{E}\left[\left\|\boldsymbol{x}^{(t,0)} - \boldsymbol{x}_i^{(t,k)}\right\|^2\right] =\eta^2 \cdot \mathbb{E}\left[\left\|\sum_{s=0}^{k-1}a_i^{(s)}g_i(\boldsymbol{x}_i^{(t,s)})\right\|^2\right] \tag{69}$$

$$\leq 2\eta^2\mathbb{E}\left[\left\|\sum_{s=0}^{k-1}a_i^{(s)}\left(g_i(\boldsymbol{x}_i^{(t,s)}) - \nabla F_i(\boldsymbol{x}_i^{(t,s)})\right)\right\|^2\right] \tag{70}$$

$$+2\eta^2\mathbb{E}\left[\left\|\sum_{s=0}^{k-1}a_i^{(s)}\nabla F_i(\boldsymbol{x}_i^{(t,s)})\right\|^2\right] \tag{71}$$

Applying Lemma 2 to the first term,

$$\mathbb{E}\left[\left\|\boldsymbol{x}^{(t,0)} - \boldsymbol{x}_i^{(t,k)}\right\|^2\right] = 2\eta^2 \sum_{s=0}^{k-1} [a_i^{(s)}]^2 \mathbb{E}\left[\left\|g_i(\boldsymbol{x}_i^{(t,s)}) - \nabla F_i(\boldsymbol{x}_i^{(t,s)})\right\|^2\right]$$

$$+ 2\eta^2 \mathbb{E}\left[\left\|\sum_{s=0}^{k-1} a_i^{(s)} \nabla F_i(\boldsymbol{x}_i^{(t,s)})\right\|^2\right] \tag{72}$$

$$\leq 2\eta^2 \sigma^2 \sum_{s=0}^{k-1} [a_i^{(s)}]^2 + 2\eta^2 \mathbb{E}\left[\left\|\sum_{s=0}^{k-1} a_i^{(s)} \nabla F_i(\boldsymbol{x}_i^{(t,s)})\right\|^2\right] \tag{73}$$

$$\leq 2\eta^2 \sigma^2 \sum_{s=0}^{k-1} [a_i^{(s)}]^2 + 2\eta^2 \left[\sum_{s=0}^{k-1} a_i^{(s)}\right] \sum_{s=0}^{k-1} a_i^{(s)} \mathbb{E}\left[\left\|\nabla F_i(\boldsymbol{x}_i^{(t,s)})\right\|^2\right] \tag{74}$$

$$\leq 2\eta^2 \sigma^2 \sum_{s=0}^{k-1} [a_i^{(s)}]^2 + 2\eta^2 \left[\sum_{s=0}^{k-1} a_i^{(s)}\right] \sum_{s=0}^{\tau_i-1} a_i^{(s)} \mathbb{E}\left[\left\|\nabla F_i(\boldsymbol{x}_i^{(t,s)})\right\|^2\right] \tag{75}$$

where (74) follows from Jensen's Inequality. Furthermore, note that

$$\frac{1}{\|\boldsymbol{a}_i\|_1} \sum_{k=0}^{\tau_i-1} a_i^{(k)} \left[\sum_{s=0}^{k-1} [a_i^{(s)}]^2\right] \leq \frac{1}{a_i} \sum_{k=0}^{\tau_i-1} a_i^{(k)} \left[\sum_{s=0}^{\tau_i-2} [a_i^{(s)}]^2\right] \tag{76}$$

$$= \sum_{s=0}^{\tau_i-2} [a_i^{(s)}]^2 = \|\boldsymbol{a}_i\|_2^2 - [a_{i,-1}]^2, \tag{77}$$

$$\frac{1}{\|\boldsymbol{a}_i\|_1} \sum_{k=0}^{\tau_i-1} a_i^{(k)} \left[\sum_{s=0}^{k-1} [a_i^{(s)}]\right] \leq \frac{1}{a_i} \sum_{k=0}^{\tau_i-1} a_i^{(k)} \left[\sum_{s=0}^{\tau_i-2} [a_i^{(s)}]\right] \tag{78}$$

$$= \sum_{s=0}^{\tau_i-2} [a_i^{(s)}] = \|\boldsymbol{a}_i\|_1 - a_{i,-1} \tag{79}$$

where $a_{i,-1}$ is the last element in the vector $\boldsymbol{a}_i$. As a result, we have

$$\frac{1}{\|\boldsymbol{a}_i\|_1} \sum_{k=0}^{\tau_i-1} a_i^{(k)} \mathbb{E}\left[\left\|\boldsymbol{x}^{(t,0)} - \boldsymbol{x}_i^{(t,k)}\right\|^2\right] \leq 2\eta^2 \sigma^2 \left(\|\boldsymbol{a}_i\|_2^2 - [a_{i,-1}]^2\right)$$

$$+ 2\eta^2 \left(\|\boldsymbol{a}_i\|_1 - a_{i,-1}\right) \sum_{k=0}^{\tau_i-1} a_i^{(s)} \mathbb{E}\left[\left\|\nabla F_i(\boldsymbol{x}_i^{(t,k)})\right\|^2\right] \tag{80}$$

In addition, we can bound the second term using the following inequality:

$$\mathbb{E}\left[\left\|\nabla F_i(\boldsymbol{x}_i^{(t,k)})\right\|^2\right] \leq 2\mathbb{E}\left[\left\|\nabla F_i(\boldsymbol{x}_i^{(t,k)}) - \nabla F_i(\boldsymbol{x}^{(t,0)})\right\|^2\right] + 2\mathbb{E}\left[\left\|\nabla F_i(\boldsymbol{x}^{(t,0)})\right\|^2\right] \tag{81}$$

$$\leq 2L^2 \mathbb{E}\left[\left\|\boldsymbol{x}^{(t,0)} - \boldsymbol{x}_i^{(t,k)}\right\|^2\right] + 2\mathbb{E}\left[\left\|\nabla F_i(\boldsymbol{x}^{(t,0)})\right\|^2\right]. \tag{82}$$

Substituting (82) into (75), we get

$$\frac{1}{\|\boldsymbol{a}_i\|_1} \sum_{k=0}^{\tau_i-1} a_i^{(k)} \mathbb{E}\left[\left\|\boldsymbol{x}^{(t,0)} - \boldsymbol{x}_i^{(t,k)}\right\|^2\right]$$

$$\leq 2\eta^2 \sigma^2 \left(\|\boldsymbol{a}_i\|_2^2 - [a_{i,-1}]^2\right) + 4\eta^2 L^2 \left(\|\boldsymbol{a}_i\|_1 - a_{i,-1}\right) \sum_{k=0}^{\tau_i-1} a_i^{(k)} \mathbb{E}\left[\left\|\boldsymbol{x}^{(t,0)} - \boldsymbol{x}_i^{(t,k)}\right\|^2\right]$$

$$+ 4\eta^2 \left(\|\boldsymbol{a}_i\|_1 - a_{i,-1}\right) \sum_{k=0}^{\tau_i-1} a_i^{(k)} \mathbb{E}\left[\left\|\nabla F_i(\boldsymbol{x}_i^{(t,0)})\right\|^2\right] \tag{83}$$

After minor rearranging, it follows that

$$\frac{1}{\|\boldsymbol{a}_i\|_1} \sum_{k=0}^{\tau_i-1} a_i^{(k)} \mathbb{E}\left[\left\|\boldsymbol{x}^{(t,0)} - \boldsymbol{x}_i^{(t,k)}\right\|^2\right] \leq \frac{2\eta^2\sigma^2}{1 - 4\eta^2 L^2 \|\boldsymbol{a}_i\|_1 (\|\boldsymbol{a}_i\|_1 - a_{i,-1})} \left(\|\boldsymbol{a}_i\|_2^2 - [a_{i,-1}]^2\right)$$

$$+ \frac{4\eta^2 \|\boldsymbol{a}_i\|_1 (\|\boldsymbol{a}_i\|_1 - a_{i,-1})}{1 - 4\eta^2 L^2 \|\boldsymbol{a}_i\|_1 (\|\boldsymbol{a}_i\|_1 - a_{i,-1})} \mathbb{E}\left[\left\|\nabla F_i(\boldsymbol{x}^{(t,0)})\right\|^2\right] \tag{84}$$

Define $D = 4\eta^2 L^2 \max_i\{\|\boldsymbol{a}_i\|_1 (\|\boldsymbol{a}_i\|_1 - a_{i,-1})\} < 1$. We can simplify (84) as follows

$$\frac{L^2}{a_i} \sum_{k=0}^{\tau_i-1} a_i^{(k)} \mathbb{E}\left[\left\|\boldsymbol{x}^{(t,0)} - \boldsymbol{x}_i^{(t,k)}\right\|^2\right] \leq \frac{2\eta^2 L^2 \sigma^2}{1-D} \left(\|\boldsymbol{a}_i\|_2^2 - [a_{i,-1}]^2\right) + \frac{D}{1-D} \mathbb{E}\left[\left\|\nabla F_i(\boldsymbol{x}^{(t,0)})\right\|^2\right]. \tag{85}$$

Taking the average across all workers and applying Assumption 3, one can obtain

$$\frac{1}{2} \sum_{i=1}^m w_i \mathbb{E}\left[\left\|\nabla F_i(\boldsymbol{x}^{(t,0)}) - \boldsymbol{h}_i^{(t)}\right\|^2\right] \leq \frac{\eta^2 L^2 \sigma^2}{1-D} \sum_{i=1}^m w_i \left(\|\boldsymbol{a}_i\|_2^2 - [a_{i,-1}]^2\right)$$

$$+ \frac{D}{2(1-D)} \sum_{i=1}^m w_i \mathbb{E}\left[\left\|\nabla F_i(\boldsymbol{x}^{(t,0)})\right\|^2\right] \tag{86}$$

$$\leq \frac{\eta^2 L^2 \sigma^2}{1-D} \sum_{i=1}^m w_i \left(\|\boldsymbol{a}_i\|_2^2 - [a_{i,-1}]^2\right)$$

$$+ \frac{D\beta^2}{2(1-D)} \mathbb{E}\left[\left\|\nabla \widetilde{F}(\boldsymbol{x}^{(t,0)})\right\|^2\right] + \frac{D\kappa^2}{2(1-D)}. \tag{87}$$

Now, we are ready to derive the final result.

### C.6 Final Results

Plugging (87) back into (64), we have

$$\frac{\mathbb{E}\left[\widetilde{F}(\boldsymbol{x}^{(t+1,0)})\right] - \widetilde{F}(\boldsymbol{x}^{(t,0)})}{\eta\tau_{\mathrm{eff}}} \leq -\frac{1}{2} \left\|\nabla \widetilde{F}(\boldsymbol{x}^{(t,0)})\right\|^2 + \tau_{\mathrm{eff}}\eta L\sigma^2 \sum_{i=1}^m \frac{w_i^2 \|\boldsymbol{a}_i\|_2^2}{\|\boldsymbol{a}_i\|_1^2}$$

$$+ \frac{\eta^2 L^2 \sigma^2}{1-D} \sum_{i=1}^m w_i \left(\|\boldsymbol{a}_i\|_2^2 - [a_{i,-1}]^2\right)$$

$$+ \frac{D\kappa^2}{2(1-D)} + \frac{D\beta^2}{2(1-D)} \mathbb{E}\left[\left\|\nabla \widetilde{F}(\boldsymbol{x}^{(t,0)})\right\|^2\right] \tag{88}$$

$$= -\frac{1}{2} \left(\frac{1 - D(1+\beta^2)}{1-D}\right) \left\|\nabla \widetilde{F}(\boldsymbol{x}^{(t,0)})\right\|^2 + \tau_{\mathrm{eff}}\eta L\sigma^2 \sum_{i=1}^m \frac{w_i^2 \|\boldsymbol{a}_i\|_2^2}{\|\boldsymbol{a}_i\|_1^2} +$$

$$+ \frac{\eta^2 L^2 \sigma^2}{1-D} \sum_{i=1}^m w_i \left(\|\boldsymbol{a}_i\|_2^2 - [a_{i,-1}]^2\right) + \frac{D\kappa^2}{2(1-D)}. \tag{89}$$

If $D \leq \frac{1}{2\beta^2+1}$, then it follows that $\frac{1}{1-D} \leq 1 + \frac{1}{2\beta^2}$ and $\frac{D\beta^2}{1-D} \leq \frac{1}{2}$. These facts can help us further simplify inequality (89).

$$
\begin{aligned}
\frac{\mathbb{E}\left[\widetilde{F}(\boldsymbol{x}^{(t+1,0)})\right] - \widetilde{F}(\boldsymbol{x}^{(t,0)})}{\eta\tau_{\text{eff}}} \leq & -\frac{1}{4}\left\|\nabla\widetilde{F}(\boldsymbol{x}^{(t,0)})\right\|^2 + \tau_{\text{eff}}\eta L\sigma^2\sum_{i=1}^{m}\frac{w_i^2\|\boldsymbol{a}_i\|_2^2}{\|\boldsymbol{a}_i\|_1^2} \\
& + \eta^2 L^2\sigma^2\left(1+\frac{1}{2\beta^2}\right)\sum_{i=1}^{m}w_i\left(\|\boldsymbol{a}_i\|_2^2 - [a_{i,-1}]^2\right) \\
& + 2\eta^2 L^2\max_i\{\|\boldsymbol{a}_i\|_1\left(\|\boldsymbol{a}_i\|_1 - a_{i,-1}\right)\}\kappa^2\left(1+\frac{1}{2\beta^2}\right) \quad (90)
\end{aligned}
$$

$$
\begin{aligned}
\leq & -\frac{1}{4}\left\|\nabla\widetilde{F}(\boldsymbol{x}^{(t,0)})\right\|^2 + \tau_{\text{eff}}\eta L\sigma^2\sum_{i=1}^{m}\frac{w_i^2\|\boldsymbol{a}_i\|_2^2}{\|\boldsymbol{a}_i\|_1^2} \\
& + \frac{3}{2}\eta^2 L^2\sigma^2\sum_{i=1}^{m}w_i\left(\|\boldsymbol{a}_i\|_2^2 - [a_{i,-1}]^2\right) \\
& + 3\eta^2 L^2\kappa^2\max_i\{\|\boldsymbol{a}_i\|_1\left(\|\boldsymbol{a}_i\|_1 - a_{i,-1}\right)\} \quad (91)
\end{aligned}
$$

Taking the average across all rounds, we get

$$
\begin{aligned}
\frac{1}{T}\sum_{t=0}^{T-1}\mathbb{E}\left[\left\|\nabla\widetilde{F}(\boldsymbol{x}^{(t,0)})\right\|^2\right] \leq & \frac{4\left[\widetilde{F}(\boldsymbol{x}^{(0,0)}) - \widetilde{F}_{\inf}\right]}{\eta\tau_{\text{eff}}T} + 4\tau_{\text{eff}}\eta L\sigma^2\sum_{i=1}^{m}\frac{w_i^2\|\boldsymbol{a}_i\|_2^2}{\|\boldsymbol{a}_i\|_1^2} \\
& + 6\eta^2 L^2\sigma^2\sum_{i=1}^{m}w_i\left(\|\boldsymbol{a}_i\|_2^2 - [a_{i,-1}]^2\right) \\
& + 12\eta^2 L^2\kappa^2\max_i\{\|\boldsymbol{a}_i\|_1\left(\|\boldsymbol{a}_i\|_1 - a_{i,-1}\right)\}. \quad (92)
\end{aligned}
$$

For the ease of writing, we define the following auxiliary variables:

$$
A = m\tau_{\text{eff}}\sum_{i=1}^{m}\frac{w_i^2\|\boldsymbol{a}_i\|_2^2}{\|\boldsymbol{a}_i\|_1^2}, \quad (93)
$$

$$
B = \sum_{i=1}^{m}w_i\left(\|\boldsymbol{a}_i\|_2^2 - [a_{i,-1}]^2\right), \quad (94)
$$

$$
C = \max_i\{\|\boldsymbol{a}_i\|_1\left(\|\boldsymbol{a}_i\|_1 - a_{i,-1}\right)\}. \quad (95)
$$

It follows that

$$
\frac{1}{T}\sum_{t=0}^{T-1}\mathbb{E}\left[\left\|\nabla\widetilde{F}(\boldsymbol{x}^{(t,0)})\right\|^2\right] \leq \frac{4\left[\widetilde{F}(\boldsymbol{x}^{(0,0)}) - \widetilde{F}_{\inf}\right]}{\eta\tau_{\text{eff}}T} + \frac{4\eta L\sigma^2 A}{m} + 6\eta^2 L^2\sigma^2 B + 12\eta^2 L^2\kappa^2 C \quad (96)
$$

Since $\min\mathbb{E}\left[\left\|\nabla\widetilde{F}(\boldsymbol{x}^{(t,0)})\right\|^2\right] \leq \frac{1}{T}\sum_{t=0}^{T-1}\mathbb{E}\left[\left\|\nabla\widetilde{F}(\boldsymbol{x}^{(t,0)})\right\|^2\right]$, we have

$$
\min_{t\in[T]}\mathbb{E}\left[\left\|\nabla\widetilde{F}(\boldsymbol{x}^{(t,0)})\right\|^2\right] \leq \frac{4\left[\widetilde{F}(\boldsymbol{x}^{(0,0)}) - \widetilde{F}_{\inf}\right]}{\eta\tau_{\text{eff}}T} + \frac{4\eta L\sigma^2 A}{m} + 6\eta^2 L^2\sigma^2 B + 12\eta^2 L^2\kappa^2 C. \quad (97)
$$

### C.7 Constraint on Local Learning Rate

Here, let us summarize the constraints on local learning rate:

$$
\eta L \leq \frac{1}{2\tau_{\text{eff}}}, \quad (98)
$$

$$
4\eta^2 L^2\max_i\{\|\boldsymbol{a}_i\|_1\left(\|\boldsymbol{a}_i\|_1 - a_{i,-1}\right)\} \leq \frac{1}{2\beta^2 + 1}. \quad (99)
$$

For the second constraint, we can further tighten it as follows:

$$4\eta^2 L^2 \max_i \{\|\boldsymbol{a}_i\|_1 (\|\boldsymbol{a}_i\|_1 - a_{i,-1})\} \le 4\eta^2 L^2 \max_i \|\boldsymbol{a}_i\|_1^2 \le \frac{1}{2\beta^2 + 1} \quad (100)$$

That is,

$$\eta L \le \frac{1}{2} \min \left\{ \frac{1}{\max_i \|\boldsymbol{a}_i\|_1 \sqrt{2\beta^2 + 1}}, \frac{1}{\tau_{\text{eff}}} \right\}. \quad (101)$$

## C.8 Further Optimizing the Bound

By setting $\eta = \sqrt{\frac{m}{\overline{\tau}T}}$ where $\overline{\tau} = \frac{1}{m} \sum_{i=1}^{m} \tau_i$, we have

$$\min_{t \in [T]} \mathbb{E} \left\| \nabla \widetilde{F}(\boldsymbol{x}^{(t,0)}) \right\|^2 \le \mathcal{O}\left( \frac{\overline{\tau}/\tau_{\text{eff}}}{\sqrt{m\overline{\tau}T}} \right) + \mathcal{O}\left( \frac{A\sigma^2}{\sqrt{m\overline{\tau}T}} \right) + \mathcal{O}\left( \frac{mB\sigma^2}{\overline{\tau}T} \right) + \mathcal{O}\left( \frac{mC\kappa^2}{\overline{\tau}T} \right). \quad (102)$$

Here, we complete the proof of Theorem 1.

## D  Proof of Theorem 2: Including Bias in the Error Bound

**Lemma 3.** *For any model parameter $\boldsymbol{x}$, the difference between the gradients of $F(\boldsymbol{x})$ and $\widetilde{F}(\boldsymbol{x})$ can be bounded as follows:*

$$\|\nabla F(\boldsymbol{x}) - \nabla \widetilde{F}(\boldsymbol{x})\|^2 \le \chi_{\boldsymbol{p}\|\boldsymbol{w}}^2 \left[ (\beta^2 - 1) \|\nabla \widetilde{F}(\boldsymbol{x})\|^2 + \kappa^2 \right] \quad (103)$$

*where $\chi_{\boldsymbol{p}\|\boldsymbol{w}}^2$ denotes the chi-square distance between $\boldsymbol{p}$ and $\boldsymbol{w}$, i.e., $\chi_{\boldsymbol{p}\|\boldsymbol{w}}^2 = \sum_{i=1}^{m} (p_i - w_i)^2 / w_i$.*

*Proof.* According to the definition of $F(x)$ and $\widetilde{F}(\boldsymbol{x})$, we have

$$\nabla F(x) - \nabla \widetilde{F}(\boldsymbol{x}) = \sum_{i=1}^{m} (p_i - w_i) \nabla F_i(\boldsymbol{x}) \quad (104)$$

$$= \sum_{i=1}^{m} (p_i - w_i) \left( \nabla F_i(\boldsymbol{x}) - \nabla \widetilde{F}(\boldsymbol{x}) \right) \quad (105)$$

$$= \sum_{i=1}^{m} \frac{p_i - w_i}{\sqrt{w_i}} \cdot \sqrt{w_i} \left( \nabla F_i(\boldsymbol{x}) - \nabla \widetilde{F}(\boldsymbol{x}) \right). \quad (106)$$

Applying Cauchy–Schwarz inequality, it follows that

$$\|\nabla F(x) - \nabla \widetilde{F}(\boldsymbol{x})\|^2 \le \left[ \sum_{i=1}^{m} \frac{(p_i - w_i)^2}{w_i} \right] \left[ \sum_{i=1}^{m} w_i \|\nabla F_i(x) - \nabla \widetilde{F}(\boldsymbol{x})\|^2 \right] \quad (107)$$

$$\le \chi_{\boldsymbol{p}\|\boldsymbol{w}}^2 \left[ (\beta^2 - 1) \|\nabla \widetilde{F}(\boldsymbol{x})\|^2 + \kappa^2 \right]. \quad (108)$$

where the last inequality uses Assumption 3. $\qquad \square$

Note that

$$\|\nabla F(\boldsymbol{x})\|^2 \le 2 \|\nabla F(\boldsymbol{x}) - \nabla \widetilde{F}(\boldsymbol{x})\|^2 + 2 \|\nabla \widetilde{F}(\boldsymbol{x})\|^2 \quad (109)$$

$$\le 2 \left[ \chi_{\boldsymbol{p}\|\boldsymbol{w}}^2 (\beta^2 - 1) + 1 \right] \|\nabla \widetilde{F}(\boldsymbol{x})\|^2 + 2\chi_{\boldsymbol{p}\|\boldsymbol{w}}^2 \kappa^2. \quad (110)$$

As a result, we obtain

$$\min_{t \in [T]} \left\| \nabla F(\boldsymbol{x}^{(t,0)}) \right\|^2 \le \frac{1}{T} \sum_{t=0}^{T-1} \left\| \nabla F(\boldsymbol{x}^{(t,0)}) \right\|^2 \quad (111)$$

$$\le 2 \left[ \chi_{\boldsymbol{p}\|\boldsymbol{w}}^2 (\beta^2 - 1) + 1 \right] \frac{1}{T} \sum_{t=0}^{T-1} \left\| \nabla \widetilde{F}(\boldsymbol{x}^{(t,0)}) \right\|^2 + 2\chi_{\boldsymbol{p}\|\boldsymbol{w}}^2 \kappa^2 \quad (112)$$

$$\le 2 \left[ \chi_{\boldsymbol{p}\|\boldsymbol{w}}^2 (\beta^2 - 1) + 1 \right] \epsilon_{\text{opt}} + 2\chi_{\boldsymbol{p}\|\boldsymbol{w}}^2 \kappa^2 \quad (113)$$

where $\epsilon_{\text{opt}}$ denotes the optimization error.

## D.1 Constructing a Lower Bound

In this subsection, we are going to construct a lower bound of $\mathbb{E}\left\|\nabla F(\boldsymbol{x})^{(t,0)}\right\|^2$, showing that (10) is tight and the non-vanishing error term in Theorem 2 is not an artifact of our analysis.

**Lemma 4.** *One can manually construct a strongly convex objective function such that* `FedAvg` *with heterogeneous local updates cannot converge to its global optimum. In particular, the gradient norm of the objective function does not vanish as learning rate approaches to zero. We have the following lower bound:*

$$\lim_{T \to \infty} \mathbb{E} \left\|\nabla F(\boldsymbol{x}^{(T,0)})\right\|^2 = \Omega(\chi^2_{\boldsymbol{p}\|\boldsymbol{w}}\kappa^2) \tag{114}$$

*where $\chi^2_{\boldsymbol{p}\|\boldsymbol{w}}$ denotes the chi-square divergence between weight vectors and $\kappa^2$ quantifies the dissimilarities among local objective functions and is defined in Assumption 3.*

*Proof.* Suppose that there are only two clients with local objectives $F_1(x) = \frac{1}{2}(x-a)^2$ and $F_2(x) = \frac{1}{2}(x+a)^2$. The global objective is defined as $F(x) = \frac{1}{2}F_1(x) + \frac{1}{2}F_2(x)$. For any set of weights $w_1, w_2, w_1 + w_2 = 1$, we define the surrogate objective function as $\widetilde{F}(\boldsymbol{x}) = w_1 F_1(\boldsymbol{x}) + w_2 F_2(\boldsymbol{x})$. As a consequence, we have

$$\sum_{i=1}^{m} w_i \left\|\nabla F_i(x) - \nabla \widetilde{F}(x)\right\|^2$$

$$= w_1[(x-a) - [x - (w_1 - w_2)a]]^2 + w_2[(x+a) - [x - (w_1 - w_2)a]]^2 \tag{115}$$

$$= w_1[2w_2a]^2 + w_2[2w_1a]^2 = 2(w_1 + w_2)(w_1w_2a^2) = 2w_1w_2a^2 \tag{116}$$

Comparing with Assumption 3, we can define $\kappa^2 = 2w_1w_2a^2$ and $\beta^2 = 1$ in this case. Furthermore, according to the derivations in Appendix A, the iterate of `FedAvg` can be written as follows:

$$\lim_{T \to \infty} x^{(T,0)} = \frac{\tau_1 a - \tau_2 a}{\tau_1 + \tau_2}. \tag{117}$$

As a results, we have

$$\lim_{T \to \infty} \left\|\nabla F(x^{(T,0)})\right\|^2 = \lim_{T \to \infty} \left[\frac{1}{2}(x^{(T,0)} - a) + \frac{1}{2}(x^{(T,0)} + a)\right]^2 \tag{118}$$

$$= \lim_{T \to \infty} \left[x^{(T,0)}\right]^2 \tag{119}$$

$$= \left(\frac{\tau_1 - \tau_2}{\tau_1 + \tau_2}\right)^2 a^2 = \frac{(\tau_2 - \tau_1)^2}{2\tau_1\tau_2}\kappa^2 = \Omega(\chi^2_{\boldsymbol{p}\|\boldsymbol{w}}\kappa^2). \tag{120}$$

where $\chi^2_{\boldsymbol{p}\|\boldsymbol{w}} = \sum_{i=1}^{m}(p_i - w_i)^2/w_i = (w_1 - 1/2)^2/w_1 + (w_2 - 1/2)^2/w_2$. $\square$

# E  Special Cases of Theorem 1

Here, we provide several instantiations of Theorem 1 and check its consistency with previous results.

## E.1  FedAvg

In `FedAvg`, $\boldsymbol{a}_i = [1, 1, \ldots, 1]^\top \in \mathbb{R}^{\tau_i}$, $\|\boldsymbol{a}_i\|_2^2 = \tau_i$, and $\|\boldsymbol{a}_i\|_1 = \tau_i$. In addition, we have $w_i = p_i\tau_i/(\sum_{i=1}^{m} p_i\tau_i)$. Accordingly, we get the closed-form expressions of the following quantities:

$$\tau_{\text{eff}} = \sum_{i=1}^{m} p_i\tau_i = \mathbb{E}_{\boldsymbol{p}}[\boldsymbol{\tau}], \tag{121}$$

$$A_{\text{FedAvg}} = m\tau_{\text{eff}} \sum_{i=1}^{m} \frac{w_i^2 \|\boldsymbol{a}_i\|_2^2}{\|\boldsymbol{a}_i\|_1^2} = \frac{m \sum_{i=1}^{m} p_i^2\tau_i}{\sum_{i=1}^{m} p_i\tau_i}, \tag{122}$$

$$B_{\text{FedAvg}} = \sum_{i=1}^{m} w_i \left(\|\boldsymbol{a}_i\|_2^2 - [a_{i,-1}]^2\right) = \frac{\sum_{i=1}^{m} p_i\tau_i(\tau_i - 1)}{\sum_{i=1}^{m} p_i\tau_i} = \mathbb{E}_{\boldsymbol{p}}[\boldsymbol{\tau}] - 1 + \frac{\text{var}_{\boldsymbol{p}}[\boldsymbol{\tau}]}{\mathbb{E}_{\boldsymbol{p}}[\boldsymbol{\tau}]}, \tag{123}$$

$$C_{\text{FedAvg}} = \max_i\{\|\boldsymbol{a}_i\|_1 (\|\boldsymbol{a}_i\|_1 - a_{i,-1})\} = \tau_{\max}(\tau_{\max} - 1). \tag{124}$$

In the case where all clients have the same local dataset size, *i.e.*, $p_i = 1/m, \forall i$. It follows that

$$\tau_{\text{eff}} = \overline{\tau}, \ A_{\text{FedAvg}} = 1, \ B_{\text{FedAvg}} = \overline{\tau} - 1 + \frac{\text{var}[\boldsymbol{\tau}]}{\overline{\tau}}, \ C_{\text{FedAvg}} = \tau_{\text{max}}(\tau_{\text{max}} - 1). \tag{125}$$

Substituting (125) into Theorem 1, we get the convergence guarantee for `FedAvg`. We formally state it in the following corollary.

**Corollary 1** (**Convergence of FedAvg**). *Under the same conditions as Theorem 1, if $p_i = 1/m$, then `FedAvg` algorithm (vanilla SGD with fixed local learning rate as local solver) will converge to the stationary point of a surrogate objective $\widetilde{F}(\boldsymbol{x}) = \sum_{i=1}^{m} \tau_i F_i(\boldsymbol{x}) / \sum_{i=1}^{m} \tau_i$. The optimization error will be bounded as follows:*

$$\min_{t \in [T]} \mathbb{E} \|\nabla \widetilde{F}(\boldsymbol{x})\|^2 \leq \mathcal{O}\left(\frac{1 + \sigma^2}{\sqrt{m\overline{\tau}T}}\right) + \mathcal{O}\left(\frac{m\sigma^2(\overline{\tau} - 1 + \text{var}[\boldsymbol{\tau}]/\overline{\tau})}{\overline{\tau}T}\right) + \mathcal{O}\left(\frac{m\kappa^2\tau_{max}(\tau_{max} - 1)}{\overline{\tau}T}\right) \tag{126}$$

*where $\mathcal{O}$ swallows all constants (including $L$), and $\text{var}[\boldsymbol{\tau}] = \sum_{i=1}^{m} \tau_i^2/m - \overline{\tau}^2$ denotes the variance of local steps.*

**Consistent with Previous Results.** When all clients perform the same local steps, *i.e.*, $\tau_i = \tau$, then $\text{var}[\boldsymbol{\tau}] = 0$ and the above error bound (126) recovers previous results [8, 24, 20]. When $\tau_i = 1$, then `FedAvg` reduces to fully synchronous SGD and the error bound (126) becomes $1/\sqrt{mT}$, which is the same as standard SGD convergence rate [51].

### E.2 FedProx

In `FedProx`, we have $\boldsymbol{a}_i = [(1-\alpha)^{\tau_i-1}, \ldots, (1-\alpha), 1]^\top \in \mathbb{R}^{\tau_i}$. Accordingly, the norms of $\boldsymbol{a}_i$ can be written as:

$$\|\boldsymbol{a}_i\|_2^2 = \frac{1 - (1-\alpha)^{2\tau_i}}{1 - (1-\alpha)^2}, \ \|\boldsymbol{a}_i\|_1 = \frac{1 - (1-\alpha)^{\tau_i}}{\alpha}, \ w_i = \frac{p_i[1 - (1-\alpha)^{\tau_i}]}{\sum_{i=1}^{m} p_i[1 - (1-\alpha)^{\tau_i}]}. \tag{127}$$

As a consequence, we can derive the closed-form expression of $\tau_{\text{eff}}, A, B, C$ as follows:

$$\tau_{\text{eff}} = \frac{1}{\alpha} \sum_{i=1}^{m} p_i[1 - (1-\alpha)^{\tau_i}], \tag{128}$$

$$A_{\text{FedProx}} = \frac{m\alpha}{\sum_{i=1}^{m} p_i(1 - (1-\alpha)^{\tau_i})} \sum_{i=1}^{m} p_i^2 \frac{1 - (1-\alpha)^{2\tau_i}}{1 - (1-\alpha)^2}, \tag{129}$$

$$B_{\text{FedProx}} = \sum_{i=1}^{m} \frac{p_i[1 - (1-\alpha)^{\tau_i}]}{\sum_{i=1}^{m} p_i[1 - (1-\alpha)^{\tau_i}]} \left[\frac{1 - (1-\alpha)^{2\tau_i}}{1 - (1-\alpha)^2} - 1\right], \tag{130}$$

$$C_{\text{FedProx}} = \frac{1 - (1-\alpha)^{\tau_{\text{max}}}}{\alpha} \left(\frac{1 - (1-\alpha)^{\tau_{\text{max}}}}{\alpha} - 1\right). \tag{131}$$

Substituting $A_{\text{FedProx}}, B_{\text{FedProx}}, C_{\text{FedProx}}$ back into Theorem 1, one can obtain the convergence guarantee for `FedProx`. Again, it will converge to the stationary points of a surrogate objective due to $w_i \neq p_i$.

**Consistency with FedAvg.** From the update rule of `FedProx`, we know that when $\mu = 0$ (or $\alpha = 0$), `FedProx` is equivalent to `FedProx`. This can also be validated from the expressions of $A_{\text{FedProx}}, B_{\text{FedProx}}, C_{\text{FedProx}}$. Using L'Hospital law, it is easy to show that

$$\lim_{\alpha \to 0} A_{\text{FedProx}} = A_{\text{FedAvg}}, \ \lim_{\alpha \to 0} B_{\text{FedProx}} = B_{\text{FedAvg}}, \ \lim_{\alpha \to 0} C_{\text{FedProx}} = C_{\text{FedAvg}}. \tag{132}$$

**Best value of $\alpha$ in FedProx.** Given the expressions of $\tau_{\text{eff}}$ and $A, B, C$, we can further select a best value of $\alpha$ that optimizes the error bound of `FedProx`, as stated in the following corollary.

**Corollary 2.** *Under the same conditions as Theorem 1 and suppose $p_i = 1/m$ and $\tau_i \gg 1$, then $\alpha = \mathcal{O}(m^{\frac{1}{2}}/\overline{\tau}^{\frac{1}{2}}T^{\frac{1}{6}}))$ minimizes the optimization error bound of `FedProx` in terms of converging to the stationary points of the surrogate objective. In particular, we have*

$$\min_{t \in [T]} \mathbb{E} \|\nabla \widetilde{F}(\boldsymbol{x})\|^2 \leq \mathcal{O}\left(\frac{1}{\sqrt{m\overline{\tau}T}}\right) + \mathcal{O}\left(\frac{1}{T^{\frac{2}{3}}}\right) \tag{133}$$

where $\mathcal{O}$ swallows all other constants. Furthermore, if we define $K = \overline{\tau}T$ the average gradient evaluations at clients and let $\overline{\tau} \leq \mathcal{O}(K^{\frac{1}{4}}m^{-\frac{3}{4}})$ (which is equivalent to $T \geq \mathcal{O}(K^{\frac{3}{4}}m^{\frac{3}{4}})$), then it follows that $\min_{t \in [T]} \mathbb{E}\|\nabla \widetilde{F}(\boldsymbol{x})\|^2 \leq \mathcal{O}(1/\sqrt{mK})$.

**Discussion:** Corollary 2 shows that there exists a non-zero value of $\alpha$ that optimizes the error upper bound of `FedProx`. That is to say, `FedProx` ($\alpha > 0$) is better than `FedAvg` ($\alpha = 0$) by a constant in terms of error upper bound. However, on the other hand, it is worth noting that the minimal communication rounds of `FedProx` to achieve $1/\sqrt{mK}$ rate, given by Corollary 2, is exactly the same as `FedAvg` [24]. In this sense, `FedProx` has the same convergence rate as `FedAvg` and cannot further reduce the communication overhead.

*Proof.* First of all, let us relax the error terms of `FedProx`. Under the assumption of $\tau_i \gg 1$, the quantities $A, B, C$ can be bounded or approximated as follows:

$$\tau_{\text{eff}} \simeq \frac{1}{\alpha}, \tag{134}$$

$$A_{\texttt{FedProx}} \simeq m\alpha \sum_{i=1}^{m} \frac{p_i^2}{(2-\alpha)\alpha} = \frac{m \sum_{i=1}^{m} p_i^2}{2-\alpha} \leq m \sum_{i=1}^{m} p_i^2 = 1, \tag{135}$$

$$B_{\texttt{FedProx}} \leq \frac{1-(1-\alpha)^{2\tau_i}}{1-(1-\alpha)^2} - 1 \leq \frac{1}{\alpha(2-\alpha)} \leq \frac{1}{\alpha} \leq \frac{1}{\alpha^2}, \tag{136}$$

$$C_{\texttt{FedProx}} \leq \frac{1}{\alpha^2}. \tag{137}$$

Accordingly, the error upper bound of `FedProx` can be rewritten as follows:

$$\min_{t \in [T]} \mathbb{E}\|\nabla \widetilde{F}(\boldsymbol{x})\|^2 \leq \mathcal{O}\left(\frac{\alpha \overline{\tau}}{\sqrt{m\overline{\tau}T}}\right) + \mathcal{O}\left(\frac{1}{\sqrt{m\overline{\tau}T}}\right) + \mathcal{O}\left(\frac{m}{\alpha^2 \overline{\tau}T}\right). \tag{138}$$

In order to optimize the above bound, we can simply take the derivative with respect to $\alpha$. When the derivative equals to zero, we get

$$\frac{\overline{\tau}}{\sqrt{m\overline{\tau}T}} = \frac{m}{\alpha^3 \overline{\tau}T} \implies \alpha = \mathcal{O}\left(\frac{m^{\frac{1}{2}}}{\overline{\tau}^{\frac{1}{2}}T^{\frac{1}{6}}}\right). \tag{139}$$

Plugging the expression of best $\alpha$ into (138), we have

$$\min_{t \in [T]} \mathbb{E}\|\nabla \widetilde{F}(\boldsymbol{x})\|^2 \leq \mathcal{O}\left(\frac{1}{\sqrt{m\overline{\tau}T}}\right) + \mathcal{O}\left(\frac{1}{T^{\frac{2}{3}}}\right) = \mathcal{O}\left(\frac{1}{\sqrt{mK}}\right) + \mathcal{O}\left(\frac{\tau^{\frac{2}{3}}}{K^{\frac{2}{3}}}\right) \tag{140}$$

where $K = \overline{\tau}T$ denotes the average total gradient steps at clients. In order to let the first term dominates the convergence rate, it requires that

$$\frac{1}{\sqrt{mK}} \geq \frac{\overline{\tau}^{\frac{2}{3}}}{K^{\frac{2}{3}}} \implies \overline{\tau} \leq \mathcal{O}\left(K^{\frac{1}{4}}m^{-\frac{3}{4}}\right). \tag{141}$$

As a results, the total communication rounds $T = K/\overline{\tau}$ should be greater than $\mathcal{O}(K^{\frac{3}{4}}m^{\frac{3}{4}})$. $\square$

# F  Proof of Theorem 3

In the case of FedNova, the aggregated weights $w_i$ equals to $p_i$. Therefore, the surrogate objective $\widetilde{F}(\boldsymbol{x}) = \sum_{i=1}^{m} w_i F_i(\boldsymbol{x})$ is the same as the original objective function $F(\boldsymbol{x}) = \sum_{i=1}^{m} p_i F_i(\boldsymbol{x})$. We can directly reuse the intermediate results in the proof of Theorem 1. According to (91), we have

$$\frac{\mathbb{E}[F(\boldsymbol{x}^{(t+1,0)})] - F(\boldsymbol{x}^{(t,0)})}{\eta\tau_{\text{eff}}} \leq -\frac{1}{4}\left\|\nabla F(\boldsymbol{x}^{(t,0)})\right\|^2 + \frac{\eta L\sigma^2 A^{(t)}}{m} + \frac{3}{2}\eta^2 L^2\sigma^2 B^{(t)} + 3\eta^2 L^2\kappa^2 C^{(t)} \tag{142}$$

where quantities $A^{(t)}, B^{(t)}, C^{(t)}$ are defined as follows:

$$A^{(t)} = m\tau_{\text{eff}} \sum_{i=1}^{m} \frac{w_i^2 \left\|\boldsymbol{a}_i^{(t)}\right\|_2^2}{\left\|\boldsymbol{a}_i^{(t)}\right\|_1^2}, \tag{143}$$

$$B^{(t)} = \sum_{i=1}^{m} p_i \left(\left\|\boldsymbol{a}_i^{(t)}\right\|_2^2 - [a_{i,-1}^{(t)}]^2\right), \tag{144}$$

$$C^{(t)} = \max_i \left\{\left\|\boldsymbol{a}_i^{(t)}\right\|_1 \left(\left\|\boldsymbol{a}_i^{(t)}\right\|_1 - a_{i,-1}^{(t)}\right)\right\}. \tag{145}$$

Taking the total expectation and averaging over all rounds, it follows that

$$\frac{\mathbb{E}[F(\boldsymbol{x}^{(T,0)})] - F(\boldsymbol{x}^{(0,0)})}{\eta\tau_{\text{eff}}T} \leq -\frac{1}{4T}\sum_{t=0}^{T-1}\mathbb{E}\left\|\nabla F(\boldsymbol{x}^{(t,0)})\right\|^2 + \frac{\eta L\sigma^2\widetilde{A}}{m}$$

$$+ \frac{3}{2}\eta^2 L^2\sigma^2\widetilde{B} + 3\eta^2 L^2\kappa^2\widetilde{C} \tag{146}$$

where $\widetilde{A} = \sum_{t=0}^{T-1} A^{(t)}/T$, $\widetilde{B} = \sum_{t=0}^{T-1} B^{(t)}/T$, and $\widetilde{C} = \sum_{t=0}^{T-1} C^{(t)}/T$. After minor rearranging, we have

$$\frac{1}{T}\sum_{t=0}^{T-1}\mathbb{E}\left[\left\|\nabla F(\boldsymbol{x}^{(t,0)})\right\|^2\right] \leq \frac{4\left[F(\boldsymbol{x}^{(0,0)}) - F_{\text{inf}}\right]}{\eta\tau_{\text{eff}}T} + \frac{4\eta L\sigma^2\widetilde{A}}{m} + 6\eta^2 L^2\sigma^2\widetilde{B} + 12\eta^2 L^2\kappa^2\widetilde{C}. \tag{147}$$

Bt setting $\eta = \sqrt{\frac{m}{\widetilde{\tau}T}}$ where $\widetilde{\tau} = \sum_{t=0}^{T-1}\overline{\tau}^{(t)}/T$, the above upper bound can be further optimized as follows:

$$\min_{t\in[T]}\mathbb{E}\left[\left\|\nabla F(\boldsymbol{x}^{(t,0)})\right\|^2\right] \leq \frac{1}{T}\sum_{t=0}^{T-1}\mathbb{E}\left[\left\|\nabla F(\boldsymbol{x}^{(t,0)})\right\|^2\right] \tag{148}$$

$$\leq \frac{4\widetilde{\tau}/\tau_{\text{eff}} \cdot \left[F(\boldsymbol{x}^{(0,0)}) - F_{\text{inf}}\right]}{\sqrt{m\widetilde{\tau}T}} + \frac{4L\sigma^2\widetilde{A}}{\sqrt{m\widetilde{\tau}T}} + \frac{6mL^2\sigma^2\widetilde{B}}{\widetilde{\tau}T} + \frac{12mL^2\kappa^2\widetilde{C}}{\widetilde{\tau}T} \tag{149}$$

$$= \mathcal{O}\left(\frac{\widetilde{\tau}/\tau_{\text{eff}}}{\sqrt{m\widetilde{\tau}T}}\right) + \mathcal{O}\left(\frac{\widetilde{A}\sigma^2}{\sqrt{m\widetilde{\tau}T}}\right) + \mathcal{O}\left(\frac{m\widetilde{B}\sigma^2}{\widetilde{\tau}T}\right) + \mathcal{O}\left(\frac{m\widetilde{C}\kappa^2}{\widetilde{\tau}T}\right). \tag{150}$$

Here, we complete the proof of Theorem 3.

Moreover, it is worth mentioning the constraints on the local learning rate. Recall that, at the $t$-th round, we have the following constraint:

$$\eta L \leq \frac{1}{2}\min\left\{\frac{1}{\max_i\left\|\boldsymbol{a}_i^{(t)}\right\|_1\sqrt{2\beta^2+1}}, \frac{1}{\tau_{\text{eff}}}\right\}. \tag{151}$$

In order to guarantee the convergence, the above inequality should hold in every round. That is to say,

$$\eta L \leq \frac{1}{2}\min\left\{\frac{1}{\max_{i\in[m],t\in[T]}\left\|\boldsymbol{a}_i^{(t)}\right\|_1\sqrt{2\beta^2+1}}, \frac{1}{\tau_{\text{eff}}}\right\}. \tag{152}$$

# G Extension: Incorporating Client Sampling

In this section, we extend the convergence guarantee of `FedNova` to the case of client sampling. Following previous works [38, 12, 20, 15], we assume the sampling scheme guarantees that the update rule (11) hold in expectation. This can be achieved by sampling with replacement from $\{1, 2, \ldots, m\}$ with probabilities $\{p_i\}$, and averaging local updates from selected clients with equal weights. Specifically, we have

$$\boldsymbol{x}^{(t+1,0)} - \boldsymbol{x}^{(t,0)} = -\tau_{\text{eff}} \sum_{j=1}^{q} \frac{1}{q} \cdot \eta \boldsymbol{d}_{l_j}^{(t)} \quad \text{where } \boldsymbol{d}_{l_j}^{(t)} = \boldsymbol{G}_{l_j}^{(t)} \boldsymbol{a}_{l_j} / \|\boldsymbol{a}_{l_j}\|_1 \tag{153}$$

where $q$ is the number of selected clients per round, and $l_j$ is a random index sampled from $\{1, 2, \cdots, m\}$ satisfying $\mathbb{P}(l_j = i) = p_i$. Recall that $p_i = n_i/n$ is the relative sample size at client $i$. For the ease of presentation, let $\boldsymbol{a}_i$ to be fixed across rounds. One can directly validate that

$$\mathbb{E}_S \left[ \frac{1}{q} \sum_{j=1}^{q} \boldsymbol{d}_{l_j}^{(t)} \right] = \frac{1}{q} \sum_{j=1}^{q} \mathbb{E}_S \left[ \boldsymbol{d}_{l_j}^{(t)} \right] = \mathbb{E}_S \left[ \boldsymbol{d}_{l_j}^{(t)} \right] = \sum_{i=1}^{m} p_i \boldsymbol{d}_i^{(t)} \tag{154}$$

where $\mathbb{E}_S$ represents the expectation over random indices at current round.

**Corollary 3.** *Under the same condition as Theorem 1, suppose at each round, the server randomly selects $q(\leq m)$ clients with replacement to perform local computation. The probability of choosing the $i$-th client is $p_i = n_i/n$. In this case,* `FedNova` *will converge to the stationary points of the global objective $F(\boldsymbol{x})$. If we set $\eta = \sqrt{q/\widetilde{\tau}T}$ where $\widetilde{\tau}$ is the average local updates across all rounds, then the expected gradient norm is bounded as follows:*

$$\min_{t \in [T]} \mathbb{E} \left\| \nabla F(\boldsymbol{x}^{(t,0)}) \right\|^2 \leq \mathcal{O} \left( \frac{\widetilde{\tau}/\tau_{\text{eff}}}{\sqrt{q\widetilde{\tau}T}} \right) + \mathcal{O} \left( \frac{\tau_{\text{eff}}/\widetilde{\tau}}{\sqrt{q\widetilde{\tau}T}} \right) + \mathcal{O} \left( \frac{q(B+C)}{\widetilde{\tau}T} \right) \tag{155}$$

*where $\mathcal{O}$ swallows all other constants (including $L, \sigma^2, \kappa^2$).*

*Proof.* According to the Lipschitz-smooth assumption, it follows that

$$\mathbb{E} \left[ F(\boldsymbol{x}^{(t+1,0)}) \right] - F(\boldsymbol{x}^{(t,0)}) \leq -\tau_{\text{eff}} \eta \underbrace{\mathbb{E} \left[ \left\langle \nabla F(\boldsymbol{x}^{(t,0)}), \sum_{j=1}^{q} \frac{\boldsymbol{d}_{l_j}^{(t)}}{q} \right\rangle \right]}_{T_3} + \frac{\tau_{\text{eff}}^2 \eta^2 L}{2} \underbrace{\mathbb{E} \left[ \left\| \sum_{j=1}^{q} \frac{\boldsymbol{d}_{l_j}^{(t)}}{q} \right\|^2 \right]}_{T_4} \tag{156}$$

where the expectation is taken over randomly selected indices $\{l_j\}$ as well as mini-batches $\xi_i^{(t,k)}, \forall i \in \{1, 2, \ldots, m\}, k \in \{0, 1, \ldots, \tau_i - 1\}$.

For the first term in (156), we can first take the expectation over indices and obtain

$$T_3 = \mathbb{E} \left[ \left\langle \nabla F(\boldsymbol{x}^{(t,0)}), \mathbb{E}_S \left[ \sum_{j=1}^{q} \frac{\boldsymbol{d}_{l_j}^{(t)}}{q} \right] \right\rangle \right] \tag{157}$$

$$= \mathbb{E} \left[ \left\langle \nabla F(\boldsymbol{x}^{(t,0)}), \sum_{i=1}^{m} p_i \boldsymbol{d}_i^{(t)} \right\rangle \right]. \tag{158}$$

This term is exactly the same as the first term in (49). We can directly reuse previous results in the proof of Theorem 1. Comparing with (56), we have

$$T_3 = \frac{1}{2} \left\| \nabla F(\boldsymbol{x}^{(t)}) \right\|^2 + \frac{1}{2} \mathbb{E} \left[ \left\| \sum_{i=1}^{m} p_i \boldsymbol{h}_i^{(t)} \right\|^2 \right] - \frac{1}{2} \mathbb{E} \left[ \left\| \nabla F(\boldsymbol{x}^{(t,0)}) - \sum_{i=1}^{m} p_i \boldsymbol{h}_i^{(t)} \right\|^2 \right] \tag{159}$$

$$\geq \frac{1}{2} \left\| \nabla F(\boldsymbol{x}^{(t)}) \right\|^2 + \frac{1}{2} \mathbb{E} \left[ \left\| \sum_{i=1}^{m} p_i \boldsymbol{h}_i^{(t)} \right\|^2 \right] - \frac{1}{2} \sum_{i=1}^{m} p_i \mathbb{E} \left[ \left\| \nabla F_i(\boldsymbol{x}^{(t,0)}) - \boldsymbol{h}_i^{(t)} \right\|^2 \right]. \tag{160}$$

For the second term in (156),

$$T_4 \leq 2\mathbb{E}\left[\left\|\frac{1}{q}\sum_{j=1}^{q}(\boldsymbol{d}_{l_j}^{(t)} - \boldsymbol{h}_{l_J}^{(t)})\right\|^2\right] + 2\mathbb{E}\left[\left\|\frac{1}{q}\sum_{j=1}^{q}\boldsymbol{h}_{l_j}^{(t)}\right\|^2\right] \tag{161}$$

$$=\frac{1}{q}\sum_{i=1}^{m}p_i\mathbb{E}\left[\left\|\boldsymbol{d}_i^{(t)} - \boldsymbol{h}_i^{(t)}\right\|^2\right] + 2\mathbb{E}\left[\left\|\frac{1}{q}\sum_{j=1}^{q}\boldsymbol{h}_{l_j}^{(t)}\right\|^2\right] \tag{162}$$

$$\leq\frac{2\sigma^2}{q}\sum_{i=1}^{m}p_i\frac{\|\boldsymbol{a}_i\|_2^2}{\|\boldsymbol{a}_i\|_1^2} + 2\mathbb{E}\left[\left\|\frac{1}{q}\sum_{j=1}^{q}\boldsymbol{h}_{l_j}^{(t)}\right\|^2\right] \tag{163}$$

$$\leq\frac{2\sigma^2}{q}\sum_{i=1}^{m}p_i\frac{\|\boldsymbol{a}_i\|_2^2}{\|\boldsymbol{a}_i\|_1^2} + 6\sum_{i=1}^{m}p_i\mathbb{E}\left[\left\|\nabla F_i(\boldsymbol{x}^{(t,0)}) - \boldsymbol{h}_i^{(t)}\right\|^2\right] + \frac{6}{q}\left(\beta^2\|\nabla F(\boldsymbol{x}^{(t,0)})\|^2 + \kappa^2\right)$$
$$+ 6\left\|\nabla F(\boldsymbol{x}^{(t,0)})\right\|^2 \tag{164}$$

where the last inequality comes from Lemma 5, stated below.

**Lemma 5.** *Suppose we are given $\boldsymbol{z}_1, \boldsymbol{z}_2, \ldots, \boldsymbol{z}_m, \boldsymbol{x} \in \mathbb{R}^d$ and let $l_1, l_2, \ldots, l_q$ be i.i.d. sampled from a multinomial distribution $\mathcal{D}$ supported on $\{1, 2, \ldots, m\}$ satisfying $\mathbb{P}(l = i) = p_i$ and $\sum_{i=1}^{m}p_i = 1$. We have*

$$\mathbb{E}[\frac{1}{q}\sum_{j=1}^{q}\boldsymbol{z}_{l_j}] = \sum_{i=1}^{m}p_i\boldsymbol{z}_i, \tag{165}$$

$$\mathbb{E}[\|\frac{1}{q}\sum_{j=1}^{q}\boldsymbol{z}_{l_j}\|^2] \leq 3\sum_{i=1}^{m}p_i\|\boldsymbol{z}_i - \nabla F_i(\boldsymbol{x})\|^2 + 3\|\nabla F(\boldsymbol{x})\|^2 + \frac{3}{q}\left(\beta^2\|\nabla F(\boldsymbol{x})\|^2 + \kappa^2\right). \tag{166}$$

*Proof.* First, we have

$$\mathbb{E}[\|\frac{1}{q}\sum_{j=1}^{q}\boldsymbol{z}_{l_j}\|^2]$$

$$=\mathbb{E}\left[\left\|\left(\frac{1}{q}\sum_{j=1}^{q}\boldsymbol{z}_{l_j} - \frac{1}{q}\sum_{j=1}^{q}\nabla F_{l_j}(\boldsymbol{x})\right) + \left(\frac{1}{q}\sum_{j=1}^{q}\nabla F_{l_j}(\boldsymbol{x}) - \nabla F(\boldsymbol{x})\right) + \nabla F(\boldsymbol{x})\right\|^2\right] \tag{167}$$

$$\leq 3\mathbb{E}[\|\frac{1}{q}\sum_{j=1}^{q}\boldsymbol{z}_{l_j} - \frac{1}{q}\sum_{j=1}^{q}\nabla F_{l_j}(\boldsymbol{x})\|^2] + 3\mathbb{E}[\|\frac{1}{q}\sum_{j=1}^{q}\nabla F_{l_j}(\boldsymbol{x}) - \nabla F(\boldsymbol{x})\|^2] + 3\|\nabla F(\boldsymbol{x})\|^2. \tag{168}$$

For the first term, by Cauchy-Schwarz inequality, we have

$$\mathbb{E}[\|\frac{1}{q}\sum_{j=1}^{q}\boldsymbol{z}_{l_j} - \frac{1}{q}\sum_{j=1}^{q}\nabla F_{l_j}(\boldsymbol{x})\|^2] \leq \frac{1}{q}\sum_{j=1}^{q}\mathbb{E}_{l_j\sim\mathcal{D}}[\|\boldsymbol{z}_{l_j} - \nabla F_{l_j}(\boldsymbol{x})\|^2] = \sum_{i=1}^{m}p_i\|\boldsymbol{z}_i - \nabla F_i(\boldsymbol{x})\|^2. \tag{169}$$

The second term can be bounded as following

$$\mathbb{E}[\|\frac{1}{q}\sum_{j=1}^{q}\nabla F_{l_j}(\boldsymbol{x}) - \nabla F(\boldsymbol{x})\|^2] = \frac{1}{q}\mathbb{E}_{l_j\sim\mathcal{D}}[\|\nabla F_{l_j}(\boldsymbol{x}) - \nabla F(\boldsymbol{x})\|^2] \tag{170}$$

$$=\frac{1}{q}\sum_{i=1}^{m}p_i\|\nabla F_i(\boldsymbol{x}) - \nabla F(\boldsymbol{x})\|^2 \tag{171}$$

$$\leq\frac{1}{q}\left[(\beta^2 - 1)\|\nabla F(\boldsymbol{x})\|^2 + \kappa^2\right]. \tag{172}$$

where the first identity follows from $\mathbb{E}_{i \sim \mathcal{D}}[F_i(\boldsymbol{x})] = \nabla F(\boldsymbol{x})$ and the independence between $l_1, \ldots, l_q$, and the last inequality is a direct application of Assumption 3.

Substituting (169) and (170) into (167) completes the proof. $\qquad\square$

Substituting (160) and (164) into (156), we have

$$
\frac{\mathbb{E}\left[F(\boldsymbol{x}^{(t+1,0)})\right] - F(\boldsymbol{x}^{(t,0)})}{\eta \tau_{\text{eff}}} \leq -\frac{1}{2}\left(1 - 6\tau_{\text{eff}}\eta L\right)\left\|\nabla F(\boldsymbol{x}^{(t,0)})\right\|^2
$$
$$
+ \left(\frac{1}{2} + 3\tau_{\text{eff}}\eta L\right) \sum_{i=1}^m p_i \mathbb{E}\left[\left\|\nabla F_i(\boldsymbol{x}^{(t,0)}) - \boldsymbol{h}_i^{(t)}\right\|^2\right]
$$
$$
+ \frac{\tau_{\text{eff}}\eta L\sigma^2}{q} \sum_{i=1}^m p_i \frac{\|\boldsymbol{a}_i\|_2^2}{\|\boldsymbol{a}_i\|_1^2} + \frac{3\tau_{\text{eff}}\eta L}{q}\left(\beta^2 \left\|\nabla F(\boldsymbol{x}^{(t,0)})\right\|^2 + \kappa^2\right)
$$
$$
\tag{173}
$$
$$
= -\frac{1}{2}\left(1 - 6\tau_{\text{eff}}\eta L - \frac{6\tau_{\text{eff}}\eta L\beta^2}{q}\right)\left\|\nabla F(\boldsymbol{x}^{(t,0)})\right\|^2
$$
$$
+ \frac{\tau_{\text{eff}}\eta L\sigma^2}{q} \sum_{i=1}^m p_i \frac{\|\boldsymbol{a}_i\|_2^2}{\|\boldsymbol{a}_i\|_1^2}
$$
$$
+ \left(\frac{1}{2} + 2\tau_{\text{eff}}\eta L\right) \sum_{i=1}^m p_i \mathbb{E}\left[\left\|\nabla F_i(\boldsymbol{x}^{(t,0)}) - \boldsymbol{h}_i^{(t)}\right\|^2\right] + \frac{3\tau_{\text{eff}}\eta L\kappa^2}{q}.
$$
$$
\tag{174}
$$

When $\eta L \leq 1/(2\tau_{\text{eff}})$ and $6\tau_{\text{eff}}\eta L + 6\tau_{\text{eff}}\eta L\beta^2/q \leq \frac{1}{2}$, it follows that

$$
\frac{\mathbb{E}\left[F(\boldsymbol{x}^{(t+1,0)})\right] - F(\boldsymbol{x}^{(t,0)})}{\eta \tau_{\text{eff}}} \leq -\frac{1}{4}\left\|\nabla F(\boldsymbol{x}^{(t,0)})\right\|^2 + \frac{\tau_{\text{eff}}\eta L\sigma^2}{q} \sum_{i=1}^m p_i \frac{\|\boldsymbol{a}_i\|_2^2}{\|\boldsymbol{a}_i\|_1^2}
$$
$$
+ \frac{3}{2} \sum_{i=1}^m p_i \mathbb{E}\left[\left\|\nabla F_i(\boldsymbol{x}^{(t,0)}) - \boldsymbol{h}_i^{(t)}\right\|^2\right] + \frac{3\tau_{\text{eff}}\eta L\kappa^2}{q}. \tag{175}
$$

Recall that the third term in (175) can be bounded as follows (see (87)):

$$
\frac{1}{2} \sum_{i=1}^m p_i \mathbb{E}\left[\left\|\nabla F_i(\boldsymbol{x}^{(t,0)}) - \boldsymbol{h}_i^{(t)}\right\|^2\right] \leq \frac{\eta^2 L^2\sigma^2}{1-D} \sum_{i=1}^m p_i\left(\|\boldsymbol{a}_i\|_2^2 - [a_{i,-1}]^2\right)
$$
$$
+ \frac{D\beta^2}{2(1-D)}\left\|\nabla F(\boldsymbol{x}^{(t,0)})\right\|^2 + \frac{D\kappa^2}{2(1-D)} \tag{176}
$$

where $D = 4\eta^2 L^2 \max_i\{\|\boldsymbol{a}_i\|_1\left(\|\boldsymbol{a}_i\|_1 - a_{i,-1}\right)\} < 1$. If $D \leq \frac{1}{12\beta^2+1}$, then it follows that $\frac{1}{1-D} \leq 1 + \frac{1}{12\beta^2} \leq 2$ and $\frac{3D\beta^2}{1-D} \leq \frac{1}{4}$. These facts can help us further simplify inequality (176). One can obtain

$$
\frac{3}{2} \sum_{i=1}^m p_i \mathbb{E}\left[\left\|\nabla F_i(\boldsymbol{x}^{(t,0)}) - \boldsymbol{h}_i^{(t)}\right\|^2\right] \leq 6\eta^2 L^2\sigma^2 \sum_{i=1}^m p_i\left(\|\boldsymbol{a}_i\|_2^2 - [a_{i,-1}]^2\right) + \frac{1}{8}\left\|\nabla F(\boldsymbol{x}^{(t,0)})\right\|^2
$$
$$
+ 12\eta^2 L^2\kappa^2 \max_i\{\|\boldsymbol{a}_i\|_1\left(\|\boldsymbol{a}_i\|_1 - a_{i,-1}\right) \tag{177}
$$
$$
= 6\eta^2 L^2\sigma^2 B + \frac{1}{8}\left\|\nabla F(\boldsymbol{x}^{(t,0)})\right\|^2 + 12\eta^2 L^2\kappa^2 C \tag{178}
$$

Substituting (178) into (175), we have

$$
\frac{\mathbb{E}\left[F(\boldsymbol{x}^{(t+1,0)})\right] - F(\boldsymbol{x}^{(t,0)})}{\eta \tau_{\text{eff}}} \leq -\frac{1}{8}\left\|\nabla F(\boldsymbol{x}^{(t,0)})\right\|^2 + \frac{\tau_{\text{eff}}\eta L\sigma^2}{q} \sum_{i=1}^m p_i \frac{\|\boldsymbol{a}_i\|_2^2}{\|\boldsymbol{a}_i\|_1^2} + \frac{3\tau_{\text{eff}}\eta L\kappa^2}{q}
$$
$$
+ 6\eta^2 L^2\sigma^2 B + 12\eta^2 L^2\kappa^2 C \tag{179}
$$
$$
\leq -\frac{1}{8}\left\|\nabla F(\boldsymbol{x}^{(t,0)})\right\|^2 + \frac{\tau_{\text{eff}}\eta L\sigma^2}{q} + \frac{3\tau_{\text{eff}}\eta L\kappa^2}{q}
$$
$$
+ 6\eta^2 L^2\sigma^2 B + 12\eta^2 L^2\kappa^2 C \tag{180}
$$

where the last inequality uses the fact that $\|\boldsymbol{a}\|_2 \leq \|\boldsymbol{a}\|_1$, for any vector $\boldsymbol{a}$. Taking the total expectation and averaging all rounds, one can obtain

$$\frac{\mathbb{E}\left[F(\boldsymbol{x}^{(T,0)})\right] - F(\boldsymbol{x}^{(0,0)})}{\eta\tau_{\mathrm{eff}}T} \leq -\frac{1}{8T}\sum_{t=0}^{T-1}\mathbb{E}\left[\left\|\nabla F(\boldsymbol{x}^{(t,0)})\right\|^2\right] + \frac{\tau_{\mathrm{eff}}\eta L(\sigma^2 + 3\kappa^2)}{q}$$
$$+ 6\eta^2 L^2\sigma^2 B + 12\eta^2 L^2\kappa^2 C. \tag{181}$$

After minor rearranging, the above inequality is equivalent to

$$\frac{1}{T}\sum_{t=0}^{T-1}\mathbb{E}\left[\left\|\nabla F(\boldsymbol{x}^{(t,0)})\right\|^2\right]$$
$$\leq \frac{8\left[F(\boldsymbol{x}^{(0,0)}) - F_{\mathrm{inf}}\right]}{\eta\tau_{\mathrm{eff}}T} + \frac{8\tau_{\mathrm{eff}}\eta L(\sigma^2 + 3\kappa^2)}{q} + 48\eta^2 L^2\sigma^2 B + 96\eta^2 L^2\kappa^2 C. \tag{182}$$

If we set the learning rate to be small enough, *i.e.*, $\eta = \sqrt{\frac{q}{\tilde{\tau}T}}$ where $\tilde{\tau} = \sum_{t=0}^{T-1}\overline{\tau}/T$, then we get

$$\frac{1}{T}\sum_{t=0}^{T-1}\mathbb{E}\left[\left\|\nabla F(\boldsymbol{x}^{(t,0)})\right\|^2\right] \leq \mathcal{O}\left(\frac{\widetilde{\tau}/\tau_{\mathrm{eff}}}{\sqrt{q\widetilde{\tau}T}}\right) + \mathcal{O}\left(\frac{\tau_{\mathrm{eff}}/\widetilde{\tau}}{\sqrt{q\widetilde{\tau}T}}\right) + \mathcal{O}\left(\frac{q(B+C)}{\widetilde{\tau}T}\right) \tag{183}$$

where $\mathcal{O}$ swallows all other constants. $\qquad\square$

## H  Pseudo-code of FedNova

Here we provide a pseudo-code of `FedNova` (see Algorithm 1) as a general algorithmic framework. Then, as an example, we show the pseudo-code of a special case of `FedNova`, where the local solver is specified as momentum SGD with cross-client variance reduction [21, 20] (see Algorithm 2). Note that when the server updates the global model, we set $\tau_{\text{eff}}$ to be the same as `FedAvg`, *i.e.*, $\tau_{\text{eff}} = \sum_{i \in \mathcal{S}_t} p_i \|\boldsymbol{a}_i^{(t)}\|_1$ where $\mathcal{S}_t$ denotes the randomly selected subset of clients. Alternatively, the server can also choose other values of $\tau_{\text{eff}}$.

---

**Algorithm 1:** FedNova Framework

**Input:** Client learning rate $\eta$; Client momentum factor $\rho$.

1 **for** $t \in \{0, 1, \dots, T-1\}$ **do**
2     Randomly sample a subset of clients $\mathcal{S}_t$
3     **Communication:** Broadcast global model $\boldsymbol{x}^{(t,0)}$ to selected clients
4     Clients perform local updates
5     **Communication:** Receive $\|\boldsymbol{a}_i^{(t)}\|_1$ and $\boldsymbol{d}_i^{(t)}$ from clients
6     Update global model: $\boldsymbol{x}^{(t+1,0)} = \boldsymbol{x}^{(t,0)} - \frac{\sum_{i \in \mathcal{S}_t} p_i \|\boldsymbol{a}_i^{(t)}\|_1}{\sum_{i \in \mathcal{S}_{\sqcup}} p_i} \sum_{i \in \mathcal{S}_t} \frac{\eta p_i \boldsymbol{d}_i^{(t)}}{\sum_{i \in \mathcal{S}_t} p_i}$
7 **end**

---

**Algorithm 2:** FedNova with Client-side Momentum SGD + Cross-client Variance Reduction

**Input:** Client learning rate $\eta$; Client momentum factor $\rho$.

1 **for** $t \in \{0, 1, \dots, T-1\}$ **at cleint $i$ in parallel do**
2     Zero client optimizer buffers $\boldsymbol{u}_i^{(t,0)} = 0$
3     **Communication:** Receive $\boldsymbol{x}^{(t,0)} = \boldsymbol{x}^{(t-1,0)} - (\sum_{i=1}^{m} p_i a_i) \eta \sum_{i=1}^{m} p_i \boldsymbol{d}_i^{(t-1)}$ from server
4     **Communication:** Receive $\sum_{i=1}^{m} p_i \boldsymbol{d}_i^{(t-1)}$ from server
5     Update gradient correction term: $\boldsymbol{c}_i^{(t)} = -\boldsymbol{d}_i^{(t-1)} + \sum_{i=1}^{m} p_i \boldsymbol{d}_i^{(t-1)}$
6     **for** $k \in \{0, 1, \dots, \tau_i - 1\}$ **do**
7        Compute: $\tilde{g}_i(\boldsymbol{x}^{(t,k)}) = g_i(\boldsymbol{x}^{(t,k)}) + \boldsymbol{c}_i^{(t)}$
8        Update momentum buffer: $\boldsymbol{u}_i^{(t,k)} = \rho \boldsymbol{u}_i^{(t,k-1)} + \tilde{g}_i(\boldsymbol{x}^{(t,k)})$
9        Update local model: $\boldsymbol{x}_i^{(t,k)} = \boldsymbol{x}_i^{(t,k-1)} - \eta \boldsymbol{u}_i^{(t,k)}$
10     **end**
11     Compute: $a_i = [\tau_i - \rho(1-\rho^{\tau_i})/(1-\rho)]/(1-\rho)$
12     Compute normalized gradient: $\boldsymbol{d}_i^{(t)} = (\boldsymbol{x}^{(t,0)} - \boldsymbol{x}^{(t,\tau_i)})/(\eta a_i)$
13     **Communication:** Send $p_i a_i$ and $p_i \boldsymbol{d}_i^{(t)}$ to the server
14 **end**

---

## I  More Experiments Details

**Platform.** All experiments in this paper are conducted on a cluster of 16 machines, each of which is equipped with one NVIDIA TitanX GPU. The machines communicate (*i.e.*, transfer model parameters) with each other via Ethernet. We treat each machine as one client in the federated learning setting. The algorithms are implemented by `PyTorch`. We run each experiments for 3 times with different random seeds.

**Hyper-parameter Choices.** On non-IID CIFAR10 dataset, we fix the mini-batch size per client as 32. When clients use momentum SGD as the local solver, the momentum factor is 0.9; when clients use proximal SGD, the proximal parameter $\mu$ is selected from $\{0.0005, 0.001, 0.005, 0.01\}$. It turns out that when $E_i = 2$, $\mu = 0.005$ is the best and when $E_i(t) \sim \mathcal{U}(2, 5)$, $\mu = 0.001$ is the best. The client learning rate $\eta$ is tuned from $\{0.005, 0.01, 0.02, 0.05, 0.08\}$ for `FedAvg` with each local solver separately. When using the same local solver, `FedNova` uses the same client learning rate as `FedAvg`.

Specifically, if the local solver is momentum SGD, then we set $\eta = 0.02$. In other cases, $\eta = 0.05$ consistently performs the best. On the synthetic dataset, the mini-batch size per client is 20 and the client learning rate is 0.02.

**Training Curves on Non-IID CIFAR10.** The training curves of `FedAvg` and `FedNova` are presented in Figure 6. Observe that `FedNova` (red curve) outperforms `FedAvg` (blue curve) by a large margin. `FedNova` only requires about half of the total rounds to achieve the same test accuracy as `FedAvg`. Besides, note that in [54], the test accuracy of `FedAvg` is higher than ours. This is because the authors of [54] let clients to perform 20 local epochs per round, which is 10 times more than our setting. In [54], after 100 communication rounds, `FedAvg` equivalently runs $100 \times 20 = 2000$ epochs.

Figure 6: Training curves on non-IID partitioned CIFAR10 dataset. In these curves, the only difference between `FedAvg` and `FedNova` is the weights when aggregating normalized gradients. 'LM' represents for local momentum. ***First row***: All clients perform $E_i = 2$ local epochs; ***Second row***: All clients perform random and time-varying local epochs $E_i(t) \sim \mathcal{U}(2, 5)$.

Figure 7: ***Left***: Comparison of different momentum schemes in `FedNova`. 'Hybrid momentum' corresponds to the combination of server momentum and client momentum. ***Right***: How `FedNova`-prox outperform vanilla `FedProx` (blue curve). By setting $\tau_{\text{eff}} = \sum_{i=1}^{m} p_i \tau_i$ instead of its default value, the accuracy of `FedProx` can be improved by $5\%$ (see the green curve). By further correcting the aggregated weights, `FedNova`-prox (red curves) achieves around $10\%$ higher accuracy than `FedProx`.