[Reviews · NeurIPS 2020]

Review 1

Summary and Contributions: The authors study the effect of the clients performing different number of local update steps in federated learning. They show that 1. simply combining the resulting updates like is done in FedAvg results in an `inconsistent' objective i.e. the algorithm ends up optimizing the wrong objective. 2. the inconsistency in the objective is reduced by using a regularization such as in FedProx, but this regularization also significantly slows down convergence 3. using normalized updates (normalized by the effective number of steps performed) corrects for this inconsistency and improves convergence.

Strengths: It was previously known that taking constant number of epochs+performing weighted averaging like is done in FedAvg is 'incorrect'. However, the prevalent wisdom was that doing so lead to better empirical performance. E.g. [R1] note that such a combination "has been shown to often outperform uniform weighting". This work closely examines this accepted wisdom and proves it wrong. I believe this is a valuable insight and adds to our understanding of federated learning. [R1]: Adaptive Federated Optimization. Reddi et al. 2020 (arxiv 2003.00295)

Weaknesses: I think some connections to existing results can be better explained. I outline two below: As the authots show, there are two factors which matter: i) the number of local update steps, and ii) the weights assigned to the updates. In standard FedAvg, the quantities p_i, w_i, and \tau_i are all proportional to n_i (number of samples). In such as case, the normalization proposed by the authors is equivalent to running FedAvg with uniform averaging. Instead of changing the averaging scheme, what happens if we simply force all workers to run the same number of updates? Secondly, the normalization proposed by the authors when using local momentum is akin to that of the "bias-correction" performed in Adam. It would have been useful to study if this actually helps when using local momentum i.e. what happens if we only normalize by the number of updates, and do not perform bias correction? ==================================================================================== I thank the authors for their answers for their responses. I agree with reviewer 4’s concern that the authors could better compare and contrast their theory with existing work. I leave my score unchanged and recommend accepting the paper, but also urge the authors to consider better and more honest discussion of what their theory explains. I thank the authors for their response to restricting the number of steps on different workers. I agree that a naive limitation would indeed slow down the convergence because we see fewer samplers and this is clearly seen in their experiment. However, it is easy to fix this by increasing the batch size on the faster workers so that the number of effective samples seen are still the same. Including experiments and discussion about this would greatly add value to the paper (even if they show FedNova performs worse/comparable to this naive scheme).

Correctness: The experiments seem valid to me, and the theorems seem intutive. I did not however examine the proofs very closely i.e. there might still be typos but I believe the results are correct.

Clarity: I found the paper well structured and easy to read.

Relation to Prior Work: The authors do an exception job of discussing and relating to prior work.

Reproducibility: Yes

Additional Feedback:


Review 2

Summary and Contributions: This paper considers the problem heterogeneity --in data distributions and nodes' computing capabilities-- in federated learning where each client runs different number of local steps in each round of the optimization routine. The paper argues that in the case that all the clients employ the same step size \eta, then the optimization iterates converge to stationary points of a surrogate function, that can be different the initial objective function. The different depends on the discrepancy among the number of local updates used by the clients. To address this issue, the authors propose FedNova in which the clients upload normalized updates to the central server and the server aggregates them according to their corresponding coefficient in the main objective.

Strengths: The main strength of this paper is providing an insight into optimization behaviors of FedAvg when dealing with different number of local updates \tau_i. It is insightful to have such results on showing how and why using the same step-sizes for clients while running different number of local updates may result in diverging from the true stationary point -- as shown in Theorem 1. It also provides a unified formulation for different cases of local solvers such as vanilla SGD (FedAvg), proximal SGD (FedProx), and momentum SGD.

Weaknesses: If we can divide the theoretical contributions of this paper into two part, they would be: 1. In Sections 3 and 4, this paper lays out the main challenge in dealing with different \tau_i's in federated optimization. In particular, Section 3 discusses a special case when the local objectives are simple linear least squares functions. Section 4 provides a general and uniform formulation for this challenge and through Theorems 1 and 2, it shows that why aggregating (unnormalized) local updates will result in convergence to a surrogate function. 2. Motivated by the result in Theorem 2, the paper proposes FedNova to correct the surrogate function back to the original objective function. That is, each client uploads normalized local updates to the central server to be aggregated. The contributions of part 1 seem marginal, although are insightful. It seems a portion of the theoretical analysis are solely spent on generalization of results of vanilla SGD to other solvers such as momentum SGD or proximal SGD; while this seems not tho be the main point of the paper. Studying different types of local solvers using heterogenous \tau_i's is certainly insightful, however in this work, it does not seem to effect the final conclusion that is the idea behind FedNova. Moreover, the paper only considers nonconvex objectives. Regarding part 2 which concerns the proposed FedNova method, it seems that the only difference with naive FedAvg is the normalization of local updates. More precisely, in FedAvg, node i uses stepsize \eta and runs SGD for \tau_i local updates and uploads \Delta_i = - \eta ( g_i(x_i^(0,t)) + ... + g_i(x_i^(\tau_i-1,t)) ) to the server, while in FedNova it uploads \Delta_i = - \tau_eff (\eta / \tau_i) ( g_i(x_i^(0,t)) + ... + g_i(x_i^(\tau_i-1,t)) ) which is the same as FedAvg's case multiplying by \tau_eff / \tau_i. In a sense, node i is now using the stepsize \tau_eff (\eta / \tau_i) rather than \eta. Although this modification resolves the inconsistency issue layed out in Theorem 1, however it is simply decreasing the affordable stepsize of each node which can degrade the convergence speed. Theorem 3 provides the main result of the proposed FedNova. It shows that if all the nodes use the same stepsize \eta = \sqrt{m / (\tilde{\tau} * T)} for all the T iterations, then the iterates of FedNova converge to stationary points of the true objective. Here \tilde{\tau} = { \bar{\tau}(1) + ... + \bar{\tau}(T) }/T is the average of all \tau_i(t) over all nodes i and all iterations t from 1 to T. This however is not feasible to employ since the values of \tau_i(t) are not realized until after each iteration, which means the value of the proposed stepsize \eta = \sqrt{m / (\tilde{\tau} * T)} is not determined till the end of the procedure. In the Appendix, a more general rate is derived in eq. (145) for generic values of \eta. There also, a condition on \eta is \eta L \leq 0.5 / \tau_eff where \tau_eff depends on \tau_i's of all the nodes; and hence the value of \tau_eff is not accessible for the nodes. Another drawback is the lack of proper comparison of FedNova with other benchmarks. Figure 2 provides simulations for FedNova and other (inconsistent) methods such as FedAvg, FedProx, etc., and shows that they are saturated around a surrogate function rather than the true objective. However, a question is how FedNova performs compared to a consistent approach. As a naive approach to address the inconsistency, each node can locally update not \tau_ times rather min{\tau_1, ... , \tau_m} times and the original stepsize \eta. That is, the slowest node determines the number of local updates. This approach also seems to enjoy the consistency property. It is not clear which of this two approaches perform better in terms of convergence rates. All in all, the proposed approach FedNova falls short in systematically addressing the heterogeneity challenge and seems to waste the higher speed of faster nodes by scaling down their stepsizes and essentially equalizing them with slower nodes.

Correctness: Theorems , 2 and 3 seem technically sound in therms of the claimed results.

Clarity: The paper is well-written and easy to follow.

Relation to Prior Work: The literature review is fairly thorough.

Reproducibility: Yes

Additional Feedback: ------- Thank you for the rebuttal. I went trough it and it has addressed some of my concerns. However, I still find the main theoretical contribution of the paper below the acceptance threshold. The main theoretical contribution of the paper (FedNova) seems marginal although I believe the detailed discussion of the objective inconsistency is insightful. Although the proposed approach -- that is basically rescaling the local updates and combining at the server -- theoretically mitigates the objective consistency, it is not clear its provable advantage over other benchmarks (FedAvg-min mentioned in my comments was a quite naive approach that comes to mind; I appreciate the response regarding this though). In general, the proposed FedNova seems too dependent to the choice of hyper-parameters.


Review 3

Summary and Contributions: - Definition of a general theoretical framework for federated learning algorithms (applicable to FedAvg and FedProx, two commonly used algorithms for Federated Learning) that allows heterogeneous number of local updates, non-IID local datasets, as well as all the generally used local solver variations. - Derivation of the convergence rates as part of this general framework providing novel insights on the convergence behavior of the previously mentioned two algorithms. - Analysis and insight into the fundamental understanding of the bias in the solution caused by objective inconsistency as a byproduct of the heterogeneous environment. - Proposal of a novel algorithm FedNova, which falls into the given general theoretical framework and compensates the induced bias whilst preserving fast convergence rates.

Strengths: S1: The paper is well structured and clearly written. S2: The proposed framework offers the possibility to gain insight on a very important problem in Federated Learning, which is the objective inconsistency given when using a simple algorithm such as FedAvg in a heterogeneous environment (different communication and computation speed between the workers). S3: Proposal of a new algorithm called FedNova based on the gained insight of the theoretical framework. Evaluation of this algorithm in terms of theoretical and empirical convergence especially compared to FedProx.

Weaknesses: W1: Thin experimental section in terms of real world data (scenario 2). Especially in terms of trained networks (only VGG-11, which is not state-of-the art in terms of accuracy nor size of the network) and datasets (only CIFAR10, which is rather small compared to other vision or NLP datasets) W2: The term normalization for the proposed algorithm FedNova is misleading. At first sight, readers might think the workers send a normalized vector (of unit length for arbitrary norm) to the centralized server, whereas in the proposed algorithm, local solvers "simply" average all the gradients for a fixed iteration t before sending this average to the centralized server. The confusion could come up if readers are aware of the existence of algorithms such as normalized gradient descent, where the solver only cares about the direction and not the magnitude of the gradient. W3: Unclear choice of the hyperparameters: From the paper it is not clear why one should use the optimized learning rate from FedAvg for FedNova as the algorithms. Additionally the hyper parameter \mu=1 for FedProx is not motivated either. Also, it is unclear, in terms of the general framework, where the difference between FedNova with a proximal local solver and FedProx lies.

Correctness: The claims and methods given in the paper seem correct. The empirical verification is rather thin. Additionally it is not clear if any scenario exists in which FedProx might outperform FedNova (and if so, in which once).

Clarity: The paper is very well structured and has a nice reading flow. The authors made the technical challenges in a heterogeneous environment clear and gave an example where simple algorithms fail. On major misunderstanding could arise from the naming of the new algorithm (see W2)

Relation to Prior Work: The relation to prior work is made clear. Especially FedProx, which was the first proposed algorithm to tackle the objective inconsistency problem, is analyzed theoretically in order to understand the slow down in convergence when compensating for the present bias. The choice of hyper parameters for this prior work is unclear.

Reproducibility: Yes

Additional Feedback: D1: The hyperparameter \mu used for the proximal version of FedNova and described in the appending has the same letter as the hyper parameter \mu=1 used for FedProx. === After rebuttal === I thank the reviewers for the clarification in the rebuttal. My concern were addressed. I believe this is insightful and important work, which is clearly motivated and well written. The experimental section could still be longer in order to clearly show that the proposed algorithm generalizes to other architecture and networks (W1). Despite this fact, I see FedNova as representing a minor contribution. The major one lies in the proposed theoretical framework and formulation of the objective inconsistency.

[Author Response · NeurIPS 2020]

Thank you to all the reviewers for the feedback! R2 and R5 are positive about the paper and gave very helpful suggestions. We believe that R4 may have a misunderstanding about the algorithm and analysis, and we have tried to clarify it below. We hope that R4 will reconsider and increase their score to recommend acceptance. Thanks again!

**[R4] Does FedNova decrease the stepsize of each client and slow down the convergence speed?** FedNova does not degrade the convergence speed of FedAvg, as we justify next. Firstly, when clients perform local updates, they use the same step size $\eta$ as FedAvg and not a scaled one $\tau_{\text{eff}}\eta/\tau_i$. Secondly, the difference between FedNova and FedAvg is the aggregation weights $w_i$, which only control the direction, and not the magnitude of the accumulated global update. The magnitude is determined by $\tau_{\text{eff}}$, which is the same in FedAvg and FedNova. For example, observe in Figure 1 that the solid green vector ($x_{\text{FedNova}}^{(t+1,0)} - x^{(t,0)}$) has roughly the same magnitude as the FedAvg update ($x_{\text{FedAvg}}^{(t+1,0)} - x^{(t,0)}$).

**[R4 & R2] What if we force all clients to run the same local steps (*e.g.*, the minimum of local steps across clients)?** It is true that forcing all clients to perform $\tau = \min_i \tau_i$ local steps (let us call this algorithm FedAvg-min) can also ensure objective consistency. However, its convergence rate is *provably worse* than FedNova. This is because, in each round, FedAvg-min will go over less data samples than FedNova ($mb\tau_{\min}$ versus $b\sum_{i=1}^{m}\tau_i$ where $b$ is the mini-batch size). Using theorem 2, one can show that the convergence rate of FedAvg-min is $1/\sqrt{mT\min_i \tau_i}$, which is slower than the rate of FedNova $1/\sqrt{T\sum_{i=1}^{m}\tau_i}$. Empirically, we evaluate the performance of FedAvg-min and FedNova on the synthetic dataset in Figure 2. Observe that FedNova achieves lower loss value than FedAvg-min at any round. Another drawback of a fixed $\tau$ algorithm like FedAvg-min is that faster nodes would remain idle in each round while waiting for slower nodes. FedNova avoids such straggling delays by allowing nodes to make different numbers of local updates.

**[R4] Do we need to know $\tilde{\tau}$ in order to choose a suitable $\eta$ in Theorem 3?** We do not need to know $\tilde{\tau}$ or $\tau_i$ beforehand. It is worth noting that $m\tilde{\tau}T = \sum_{i=1}^{m}\sum_{t=0}^{T-1}\tau_i^{(t)}$ is actually the total number of processed mini-batches across all clients after $T$ rounds. Once we have a budget on the total mini-matches $K = m\tilde{\tau}T$ to be processed, we can set the learning rate as $\eta = \sqrt{m/\tilde{\tau}T} = \sqrt{m^2/K}$, then the optimization error is guaranteed to be bounded by $\mathcal{O}(1/\sqrt{K})$. As for the upper bound on learning rate, it is only used for theoretical analysis. In practice, one always needs to tune the learning rate.

Novel Generalized Update Rule

$$x^{(t+1,0)} = x^{(t,0)} - \tau_{\text{eff}}\sum_{i=1}^{m}w_i \cdot \eta d_i^{(t)}$$

Optimizes $\widetilde{F}(x) = \sum_{i=1}^{m}w_i F_i(x)$

Figure 1: The difference between FedAvg and FedNova is the aggregation weights $w_i$, which only controls the direction of the solid green arrow.

Figure 2: FedAvg-min.

**[R4] Is the general analysis framework a marginal contribution?** We believe that the analytical framework proposed in Section 4 is an important and impactful contribution, perhaps even more critical than Section 5. This is because: 1) we identify the objective inconsistency problem in FedAvg by showing that performing the same number of local epochs at clients with heterogeneous sizes datasets optimizes a mismatched objective and 2) we provide the first (to the best of our knowledge) rigorous understanding of the objective inconsistency problem in federated learning by quantifying the non-vanishing gap caused by incorrect weighted aggregation of heterogeneously updated models.

**[R4] Extending our theorems to strongly convex case.** We focus on the non-convex case since it is the most practical and challenging setting. It is straightforward to extend our analysis to convex or strongly convex cases. For instance, one can directly apply Polyak-Łojasiewicz condition to Eqn. (89) in the appendix and obtain an improved rate.

**[R2] Is the bias-correction necessary when using local momentum?** Yes, it is necessary because without bias-correction, the algorithm will converge to a stationary point of a mismatched objective, the analytical form of which can be derived using our framework. We will add some experiments in appendix to further validate this.

**[R5] Clarifications on FedProx: hyper-parameters and differences to FedNova.** On the synthetic dataset, we use the same model and hyper-parameters as the FedProx paper. We set $\mu = 1$ because this is the best value reported in that paper. On the CIFAR-10 dataset, we tuned the value of $\mu$ from $\{0.0005, 0.001, 0.005, 0.01\}$ as stated in the Appendix. FedNova with proximal updates is same as FedProx in terms of the local updates, but the aggregation weights $w_i$ and effective steps $\tau_{\text{eff}}$ are set differently. In our framework (4), the weights and $\tau_{\text{eff}}$ used in FedProx are given by Eqn. (6) while FedNova uses $w_i = p_i$ and $\tau_{\text{eff}} = \sum_{i=1}^{m}p_i\tau_i$.

**[R5] Avoiding confusions on the algorithm name.** Thanks for the suggestion! We will avoid using the term 'normalized gradient' and clearly state the meaning of normalization in our paper, or even use another term.

[Meta-Review · NeurIPS 2020]

The paper studies federated learning, when agents perform different number of local update steps. It shows that normalizing these updates by the effective number of steps performed allows to converge to the correct objective value (and incorrect without such a normalization). Both are valuable new insights to federated learning, and important in practice. Some concerns remained on clarifying the positioning with respect to related work, as well as hyperparameter choices, but overall consensus was positive. We hope the exceptionally detailed feedback with improvement suggestions from the 3 reviews will be implemented for the camera ready version.